# Targeted degradation of extracellular mitochondrial aspartyl-tRNA synthetase modulates immune responses

Benjamin S. Johnson[1], Daniela Farkas[1], Rabab El-Mergawy[1], Jessica A. Adair[1], Ajit Elhance [1], Moemen Eltobgy[1], Francesca M. Coan[1], Lexie Chafin[1], Jessica A. Joseph[1], Alex Cornwell [1], Finny J. Johns[1], Lorena Rosas[1], Mauricio Rojas[1], Laszlo Farkas [1], Joseph S. Bednash[1], James D. Londino [1], Prabir Ray [2], Anuradha Ray [2], Valerian Kagan [3], Janet S. Lee [4], Bill B. Chen [2] & Rama K. Mallampalli [1] ✉

The severity of bacterial pneumonia can be worsened by impaired innate immunity resulting in ineffective pathogen clearance. We describe a mitochondrial protein, aspartyl-tRNA synthetase (DARS2), which is released in circulation during bacterial pneumonia in humans and displays intrinsic innate immune properties and cellular repair properties. DARS2 interacts with a bacterial-induced ubiquitin E3 ligase subunit, FBXO24, which targets the synthetase for ubiquitylation and degradation, a process that is inhibited by DARS2 acetylation. During experimental pneumonia, Fbxo24 knockout mice exhibit elevated DARS2 levels with an increase in pulmonary cellular and cytokine levels. In silico modeling identified an FBXO24 inhibitory compound with immunostimulatory properties which extended DARS2 lifespan in cells. Here, we show a unique biological role for an extracellular, mitochondrially derived enzyme and its molecular control by the ubiquitin apparatus, which may serve as a mechanistic platform to enhance protective host immunity through small molecule discovery.

Despite the emergence of potent antibiotics, bacterial pneumonia remains a leading cause of infectious deaths worldwide with over two and one half million deaths yearly[1]. The pathobiology of both hospital-acquired and community acquired bacterial pneumonia usually involves microaspiration of oral or nasopharyngeal pathogens that triggers a complex and highly orchestrated array of host immune responses. These responses involve the induction of innate immune pathways with the release of cytokines and chemokines by resident cells within the lung microenvironment, effector cell recruitment of monocytes/macrophages and polymorphonuclear cells, and engagement of T and B lymphocytes of adaptive immune pathways that are

integral to pathogen eradication, cellular repair, and resolution of lung inflammation. In the absence of an appropriate immune response or effective anti-microbial treatment, patients with lower respiratory tract infections may develop collateral tissue damage or lung injury, clinically termed the acute respiratory distress syndrome (ARDS), or develop sepsis, disorders both characterized by immune dysregulation and, in some circumstances, possess features of immune-suppression[2,3]. Emerging data has identified an immuno-suppressed state in some sepsis patients that also exists in patients with pneumonia characterized by diminished host protective cytokines (e.g. TNF-α, IL-1β, IL-6, and IL-10), and increased release of the

[1]Department of Internal Medicine, Division of Pulmonary, Critical Care, and Sleep Medicine, The Ohio State University, Columbus, OH, USA. [2]Department of Medicine, Division of Pulmonary, Allergy, and Critical Care Medicine, the University of Pittsburgh, Pittsburgh, PA, and Sleep Medicine, Pittsburgh, PA, USA. [3]Department of Environmental and Occupational Health, University of Pittsburgh, Pittsburgh, PA, USA. [4]Division of Pulmonary and Critical Care Medicine, Department of Medicine, Washington University, St. Louis, MO, USA. ✉e-mail: rama.mallampalli2@osumc.edu

immunosuppressive IL-1 receptor antagonist[4]. Given the defects in host immune responses and the emergence of antibiotic-resistant bacterial strains in pneumonia, discovery of newer cytoprotective and innate immune networks is essential for development of non-antibiotic therapeutics that target these pathways.

Mitochondrial aspartyl-tRNA synthetase (DARS2), a member of a heterogenous family of mitochondrial aminoacyl-tRNA synthetase proteins (mt-aaRSs), is a tRNA ligase responsible for conjugation of aspartate with its respective tRNA making it an essential regulator of mitochondrial biogenesis[5]. The indispensability of DARS2 is evident from its targeted disruption, which is embryonically lethal in mice[6]. DARS2 has multiple naturally occurring point mutations that cause leukoencephalopathy with brainstem and spinal cord involvement and lactate elevation (LBSL)[7,8]. LBSL is a rare, but aggressive neurodegenerative disorder in which atrophy of the white matter tracks and impaired myelin turnover result in motor impairment, sensory loss and is potentially fatal[7,8]. Several common disease-causing mutations of DARS2 have been found to alter its biology through disruption of dimerization, catalytic activity, or reduced expression levels[9]. In addition to its known contribution to LBSL, DARS2 has been implicated as an oncogene in hepatocellular carcinoma and lung adenocarcinoma[10–13]. Recent data suggest that DARS2 may play a vital role in macrophage-dependent wound healing and immune cell recruitment in lung adenocarcinoma[14–16]. To date, however, knowledge of the molecular regulation of DARS2 is limited. Ubiquitylation is a fundamental modification for protein degradation that modulates immune processes. Ubiquitylation requires ligation of a substrate to ubiquitin via E3 ligase complexes as the last step of the E1-E2-E3 ubiquitylation cascade. Hundreds of E3 ligases exist naturally, however, much of their molecular behavior and biological roles remain undefined. The Skp-Cullin-1-F-Box (SCF) E3-ligase family through its substrate recognizing subunit, termed F-box protein, has emerged as a key regulator of diverse processes including transcriptional control, cellular proliferation, neoplasia, and immunity[17]. Recent identification of E3 ligases, F-box proteins, and their putative targets has led to a pipeline of small molecules that have the potential to transition as a new class of therapeutics in the clinic for malignancies[18,19].

Here we discovered that the E3 ligase component, FBXO24, is an immunoregulatory protein via the cellular disposal of DARS2, a previously unrecognized secreted protein in human plasma with cytoprotective and immune properties. Based on the DARS2-FBXO24 molecular interaction from virtual screening, we identified an FBXO24 inhibitor compound, that when administered in vivo, displays immunostimulatory activities.

## Results

### FBXO24 is induced by bacterial infection

In the process of studying ubiquitin associated proteins in pneumonia, human transplant rejected lung tissue testing positive in bacterial cultures had a significant increase in FBXO24 mRNA and protein levels compared to uninfected tissue or to other related F-box proteins (Fig. 1a–c, Supplementary Fig. 1). For confirmation, BEAS-2B and A549

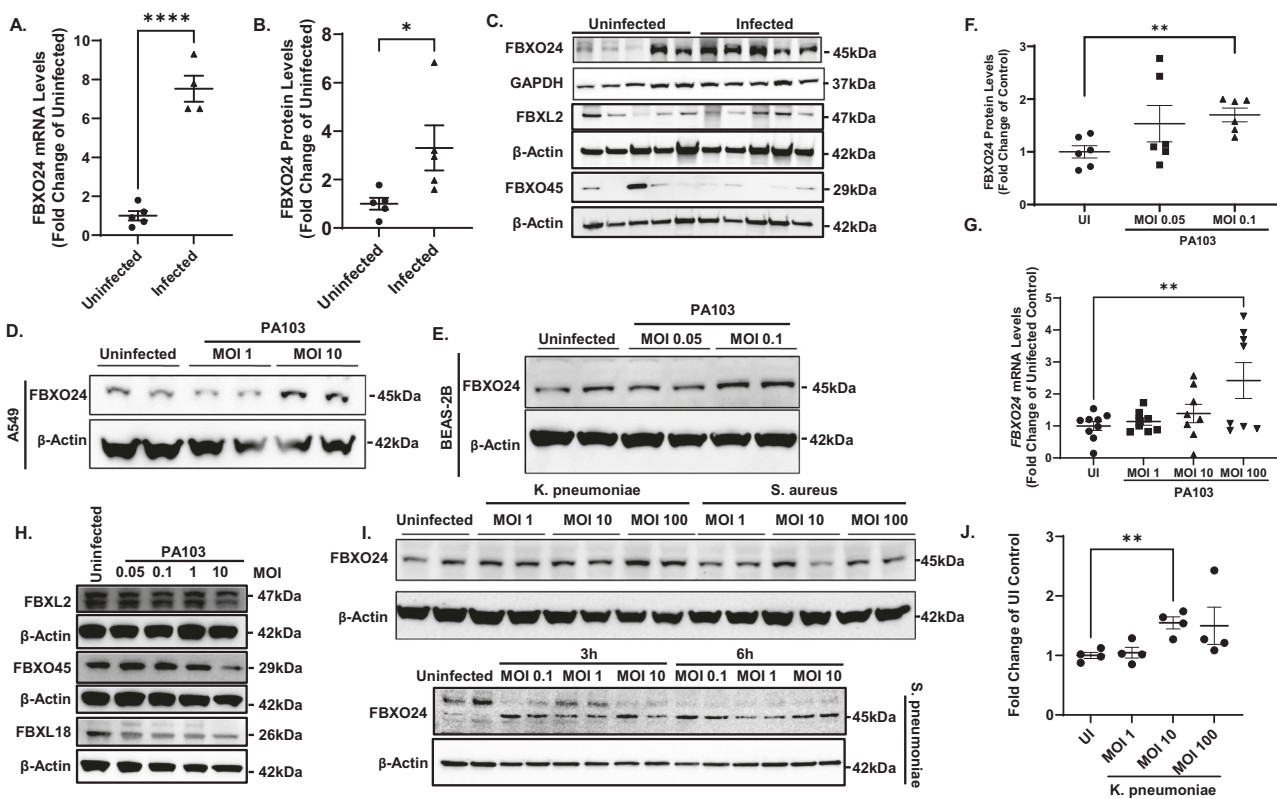

**Fig. 1 | FBXO24 is induced by bacterial infection. A** mRNA expression of FBXO24 in infected (confirmed by positive culture) transplant rejected human lung tissue measured by RT-qPCR, $n = 4$-5 patient samples ($p < 0.0001$) and (**B**) levels of FBXO24 protein quantitated in infected vs. uninfected tissue $n = 5$ patient samples/group, ($p = 0.0434$). **C** Representative immunoblots showing changes in F-box proteins from transplant rejected human lung tissue. **D, E** Representative immunoblots of FBXO24 protein levels in A549 (**D**) and BEAS2B (**E**) cells 6 h after infection with PA103. **F** Quantification by densitometry of panel (**E**) and additional samples ($p = 0.0024$) $n = 6$ biologically independent samples. **G** mRNA levels of FBXO24 in BEAS2B cells after 6 h PA103 infection, $n = 8$-9 samples/group, ($p = 0.0095$). **H** Shown are levels of related F-box proteins in BEAS2B cells after 6 h PA103 infection. **I** FBXO24 protein following 6 h *Klebsiella pneumoniae* or *Staphylococcus aureus* infection in BEAS-2B (top blot) and *Streptococcus peumoniae* at 3 or 6 h. **J** Quantification of FBXO24 protein in BEAS-2B infected with *Klebsiella pneumoniae* from panel (**I**) and additional samples. $n = 4$ biologically independent samples, ($p = 0.0029$). **A, B, F, G,** and **J** data are presented as mean values +/- SEM. **A, B, F** Unpaired student's t-test, two tailed. **G, J** One-way ANOVA with multiple comparisons. Source data are provided as a Source Data file.

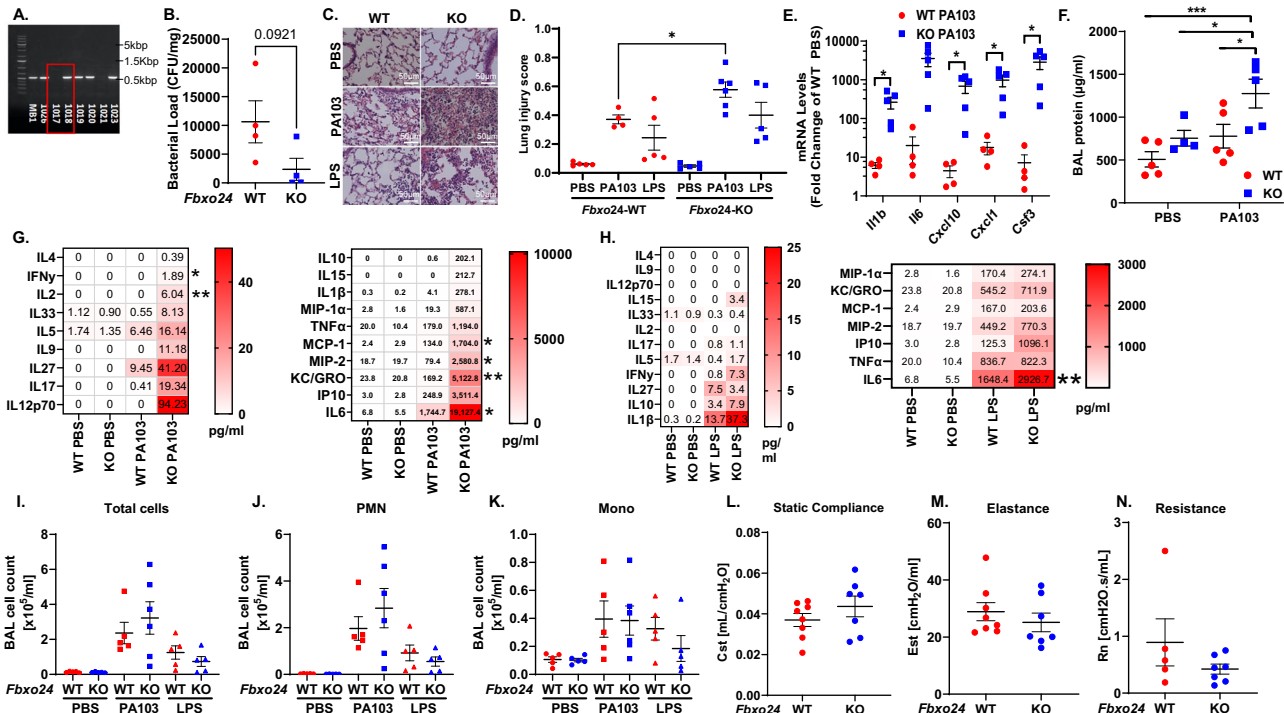

**Fig. 2 | Fbxo24 genetic ablation in mice increases innate immunity. A** Detection of *Fbxo24* gene deletion via qPCR in founder mice. **B**–**D** Wild-type (WT) littermates or *Fbxo24* KO mice were given PA103 (5×10⁵ cfu/mouse or LPS (3 mg/kg) intranasally (i.n.) and 24 h later mice were euthanized, lungs lavaged, lung tissue harvested for analysis of bacterial load **B**, lung histology **C** and lung injury score, (p = 0.0192) **D**. **E** Lung tissue cytokine mRNA analysis via RT-qPCR from PA103 (5 × 10⁵ cfu/mouse) infected WT vs. *Fbxo24* KO mice (Il1b p = 0.0434, Cxcl10 *p* = 0.0467, Cxcl1 *p* = 0.0401, Csf3 *p* = 0.0495). **F** BAL protein from PA103 infected WT vs. *Fbxo24* KO mice (from top *p* = 0.0018, *p* = 0.0395, *p* = 0.037). **G, H** Cytokine levels in the BAL of WT vs. *Fbxo24* KO mice infected with PA103 (IFNy p = 0.0154, IL-2 p = 0.0094, MCP-1 *p* = 0.0157, MIP-2 *p* = 0.0089, KC/GRO *p* = 0.0149, IL-6 *p* = 0.01 for WT PA103 vs. KO PA103) **G** or treated with LPS (IL-6 *p* < 0.0001 for WT PA103 vs. KO PA103) **H** and

PBS controls assayed via multiplex ELISA. **I** BAL total cell counts, **J** PMN and **K** mononuclear (Mono) cell levels from PA103 infected and LPS treated WT vs. *Fbxo24* KO mice. **L**–**N** WT and *Fbxo24* KO mice given PA103 (1 × 10⁵ cfu/mouse) i.n. were analyzed for lung mechanics showing no change in compliance **L**, elastance **M**, or resistance **N**. B n = 4 mice/group and **D**–**K** n = 4–6 mice/group. **L**–**N** *n* = 7 (KO) *and* 8 (WT)/group. B, **D**–**F**, **I**–**N** data are presented as mean values ± SEM. B, D, E Unpaired student's t-test, two-tailed. F One-way ANOVA with Dunnett's multiple comparison test. G One-way ANOVA with Tukey's multiple comparison test used for statistical analysis, applied to each cytokine's data individually. **H** *P*-values derived multiple unpaired two-tailed student's t-tests. Source data are provided as a Source Data file.

cells were infected with the *P. aeruginosa* strain PA103 (MOI 0.05-100), and both showed increases in immunoreactive FBXO24 (Fig. 1d–f), an effect associated with increases in FBXO24 mRNA (Fig. 1g). PA103 did not significantly induce levels of other tested F-box proteins (Fig. 1h). Likewise, unlike *S. aureus*, the bacterial pathogens *K. pneumoniae, S. pneumoniae*, and *E. coli* infection all increased FBXO24 protein expression in BEAS-2B cells or THP-1 cells (Fig. 1i, j, Supplementary Fig. 1d,e). These data suggest that FBXO24 levels are variably induced depending on the pathogen or cell type. Finally, in assessing biologic networks regulated by FBXO24, we observed that transcriptomic studies uncovered several pathways impacted by cellular knockdown of the F-box protein including metabolic, proliferative, and checkpoints involved in cellular energetics and innate immunity (Supplementary Fig. 2).

## Fbxo24 genetic ablation in mice enhances innate immunity

To unravel the biological relevance of Fbxo24, we generated a breeding colony of *Fbxo24* knockout (KO) and wild type (WT) mice. *Fbxo24* gene deletion was confirmed via PCR amplification and agarose gel analysis (Fig. 2a). Mice were given PA103 (5×10⁵ cfu/mouse), lipopolysaccharide (LPS), or vehicle (PBS) intranasally (i.n.) and 24 h later animals were euthanized, bronchoalveolar lavage (BAL) fluid was collected, and tissues processed for analysis. *Fbxo24* KO mice showed a trend toward decreased bacterial loads compared to WT mice (Fig. 2b *P* = 0.09). Lung histological analysis indicated that *Fbxo24*-KO mice infected with PA103 had increased parenchymal cell infiltration and a

significant increase in lung injury score compared to PA103 infected WT mice (Fig. 2c, d). *Fbxo24* KO mice displayed increased expression of *Il1b, Cxcl10, KC* and *CSF3* gene expression compared to WT controls when infected with PA103 (Fig. 2e). Consistent with transcript induction, PA103 infected *Fbxo24* KO mice showed increased lung microvascular permeability as evidenced by elevated BAL protein, and levels of several proinflammatory mediators (IL-2, IL-5, IL-6, MCP-1, MIP-2 and KC/GRO) compared to infected WT mice (Fig. 2f, g). LPS treated *Fbxo24* KO mice also had a significant increase in IP10 levels in BAL and sizeable increases in other proinflammatory mediators (IL-6, MCP-1, MIP-2 and KC/GRO). However, these differences did not reach statistical significance (Fig. 2h). BAL cell populations were not altered in WT versus *Fbxo24* KO mice significantly after PA103 infection or LPS challenge (Fig. 2i–k). Given the robust increase in tissue injury and inflammation in *Fbxo24* KO mice, we evaluated the effect of bacterial infection on lung function. *Fbxo24* WT and *Fbxo24* KO mice infected i.n. with PA103 (1 × 10⁵ CFU) displayed no significant difference in lung mechanics (Fig. 2l–n). Here we used a lower amount of bacterial infection in mice because of technical challenges using the Flexivent system using viable mice to measure lung function. Nevertheless, under these conditions the data suggest that Fbxo24 depletion triggers increased lung innate immune activity that is not associated with significant impairment of lung mechanics. As a complementary model to assess Fbxo24 on systemic inflammation, mice were intraperitoneally (i.p.) injected with LPS or vehicle (Supplementary Fig. 3). *Fbxo24* KO mice challenged with LPS had statistically significant

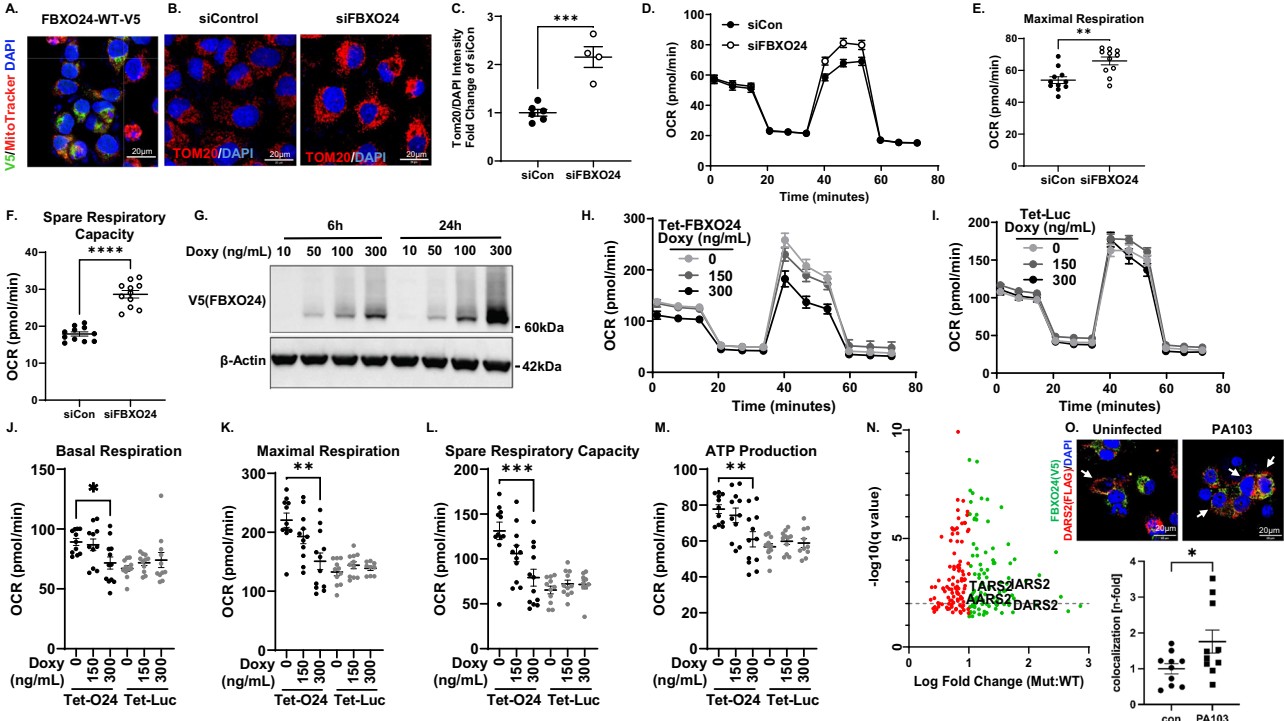

**Fig. 3 | FBXO24 impairs mitochondrial respiration. A** A549 cells were transfected with V5-Tagged *FBXO24* and stained for V5, MitoTracker Deep Red, and DAPI showing sub-cellar localization. **B, C** Representative images of A549 cells transfected with control or *FBXO24* siRNA and stained for TOM20 and DAPI showing increased TOM20 suggestive of increased mitochondrial signals as seen in **C**, (*p* = 0.0003, *n* = 4–6 biologically independent samples/group). **D–F** BEAS-2B cells transfected with 10 nM control RNA or *FBXO24* siRNA for 48 h and oxygen consumption ratio (OCR) **D**, maximal respiration, (*p* = 0.0014) **E**, and spare respiratory capacity, (*p* < 0.0001), **F** was assayed using a Seashore XFe96 Analyzer. *n* = 11 biologically independent samples/group; repeated 3 times and representative experiment shown. **G** Confirmation of FBXO24 induction using varying doxycycline (Doxy) concentrations in BEAS2B-Tet-FBXO24 cells (Tet-O24). Representative plots in oxygen consumption **H** Tet-O24 cells or a control cell line (Tet-Luc, [luciferase]) in **I** where Beas2Bs-Tet-Luc cells were treated with vehicle, 150 ng/ml or 300 ng/mL doxy overnight. Tet-O24 cells demonstrated dose dependent changes in **J** basal respiration, (*p* = 0.0163), **K** maximal respiration, (0.0012,), **L** spare respiratory capacity, (p = 0.0009), and **M** ATP production,

(*p* = 0.0059). *N* = 10–11 biological replicates per group; repeated 2 times, representative data shown. **N** A549 cells stably over-expressing FBXO24-Wt MiniTurboID or FBXO24-LPAA-MiniTurboID were treated with biotin (4 h) prior to lysis. Biotinylated proteins were captured using streptavidin beads and submitted for mass spectrometry. Shown are peptides that were significantly upregulated in FBXO24 and FBXO24-LP/AA mutant cells vs. mNEON controls. We compared peptides significantly upregulated after inactivation of the FBXO24 activity by measuring peptide abundance in FBXO24 vs. FBXO24-LP/AA expressing cells. Green > 1-fold change vs. FBXO24. Red <1 fold change vs. FBXO24 (*n* = 5 biologically independent samples/group). **O** Shown is subcellular localization of ectopically expressed Flag-Tagged-*DARS2* and V5-tagged-*FBXO24* in A549 cells infected with PA103 or uninfected. Scale bar=20 µm. Representative immunofluorescence images and quantification (*p* = 0.0447, *n* = 10 biological replicates/group) of DARS2-FBXO24 colocalization (arrows) are shown below. **C–F, H–M** and **O** data are presented as mean values±SEM. **C, E, F, O** p-values derived from unpaired student's t-test. **J–M** Analyzed with One-way ANOVA with multiple comparisons. Source data are provided as a Source Data file.

increases in *Cxcl10 and Ccl5* lung mRNA and liver *Cxcl10* and *Il1β* mRNA (Supplementary Fig. 3a, b). LPS challenged *Fbxo24* KO mice also displayed significantly higher levels of BAL cells including PMNs and mononuclear cells as well as circulating IL-6 in the plasma compared to LPS treated WT controls (Supplementary Fig. 3c, d). Lastly, *Fbxo24* KO mice displayed modestly increased DARS2 protein in lung tissue compared to WT controls (Supplementary Fig. 3e). Hence, Fbxo24 KO mice have a striking increase in inflammatory responses with attendant lung injury after bacterial infection suggesting its role in suppressing innate immune pathways.

## FBXO24 impairs mitochondrial function
We assessed FBXO24 mechanisms first focusing on mitochondrial function, as changes in energetics were observed in *FBXO24* depletion transcriptomic studies and these organelles are linked to adequate cytokine synthesis[20]. Ectopically expressed V5-tagged-*FBXO24* in A549 cells showed colocalization of the F-box protein with MitoTracker Red indicating mitochondrial localization (Fig. 3a). *FBXO24* siRNA treated A549 cells showed significantly increased intensity of TOM20 fluorescence in cells when normalized to DAPI

counterstaining (Fig. 3b, c) compared to control siRNA. Knockdown of *FBXO24* in BEAS-2B cells significantly increased maximal respiration and spare respiratory capacity compared to controls (Fig. 3d–f). Doxycycline-inducible overexpression of *FBXO24* (Tet-FBXO24) in stably integrated BEAS-2B cells, unlike an inducible reporter (luciferase, Tet-Luc) control, showed decreased mitochondrial respiration (Fig. 3g–i). Further, doxycycline-inducible overexpression of *FBXO24* produced a significant dose dependent suppression of basal, maximal, and spare respiratory capacity and decreased ATP production (Fig. 3j–m). We next investigated possible molecular targets within mitochondria that might mediate alterations in immune responses and bioenergetics by FBXO24. Proximity dependent biotinylation assays using wild type FBXO24 or catalytically inactive FBXO24 (FBXO24-LPAA) as a bait conjugated with TurboID identified DARS2 and several mt-aaRS family members as having increased labeling in the FBXO24-LPAA group (Fig. 3n). The data strongly suggested that FBXO24-mt-aaRS interaction may be dependent on SCF[FBXO24] E3-ligase activity. In agreement with this data, FBXO24 and DARS2 colocalized, an effect enhanced after cellular PA103 infection (Fig. 3o).

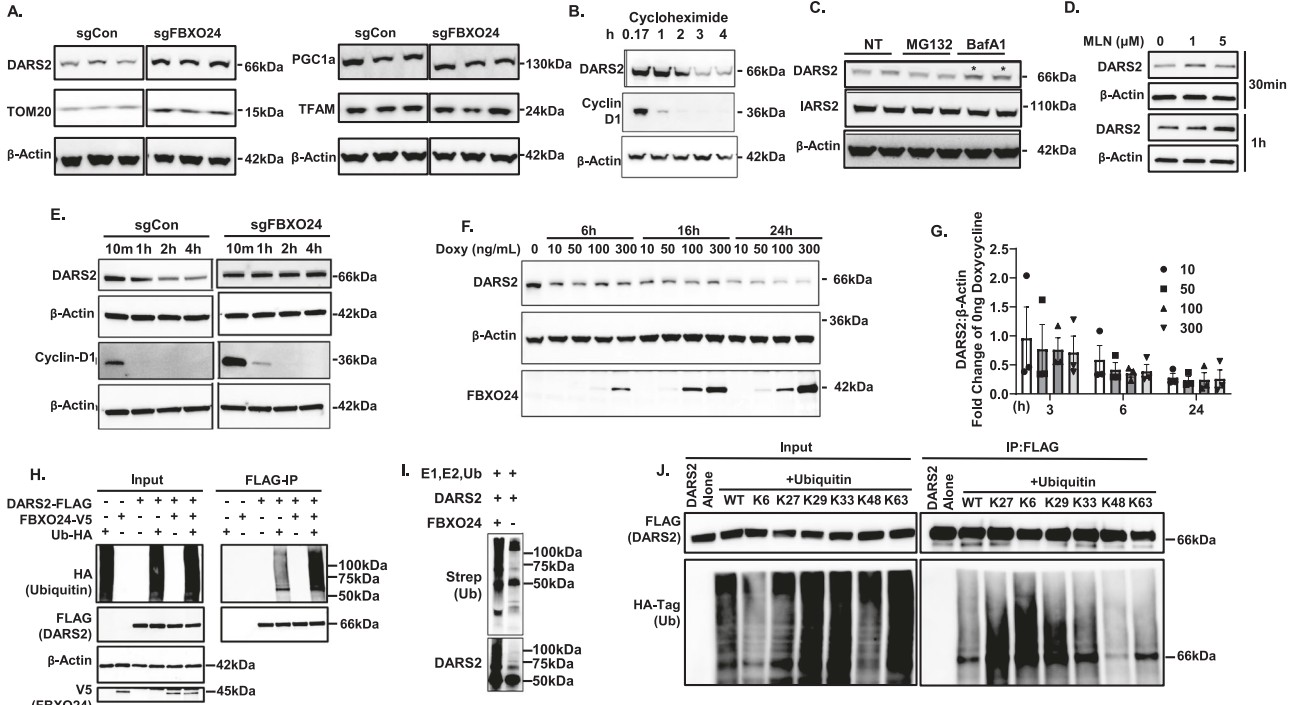

**Fig. 4 | FBXO24 binds and mediates DARS2 ubiquitylation and degradation.**
**A** Levels of mitochondrial biogenesis proteins in a CRISPR generated *FBXO24* KO BEAS-2B cell line, and a control cell line, where higher levels of DARS2 and TOM20 proteins are shown ($n = 2$). Each lane shown is a separate sample. **B** DARS2 t ½ determined using cycloheximide (CHX) (40μg/mL) treatment for various times in BEAS-2B cells. Shown is a representative immunoblot using cyclin D as a control. **C** DARS2 and IARS2 levels by immunoblotting in BEAS-2B cells treated with MG-132 [10 μM] proteasome inhibitor or bafilomycin A1 (BafA1)[100 nM] lysosome inhibitor ($n = 3$). **D** Immunoreactive DARS2 levels in BEAS2B cells treated with vehicle (0), or a pan ubiquitination inhibitor MLN (1 or 5 μM) for 30 min or 1 h ($n = 3$). **E** DARS2 t ½ in sg*Control* or sg*FBXO24* BEAS-2B cells treated with CHX up to 4 h with a representative immunoblot ($n = 3$). **F** Effect of overexpressed *FBXO24* on

endogenous DARS2 levels in doxy-inducible FBXO24 BEAS-2B cells, representative immunoblot and **G** quantification (repeated $n = 3$ times) Data are presented as mean values±SEM. **H** Polyubiquitylation levels of ectopically expressed Flag-tagged-*DARS2* in the presence or absence of *FBXO24*-V5 or HA-tagged-*ubiquitin* (Ub). Cells were transfected with plasmids and processed for immunoprecipitation (IP) and immunoblotting ($n = 3$). **I** In vitro ubiquitination reaction of DARS2 with or without FBXO24 showing polyubiquitylated DARS2 ($n = 1$). **J** IP of ectopically expressed Flag-*DARS2* co-transfected with WT or R→K HA-*ubiquitin* constructs to identify primary ubiquitin linkages ($n = 3$). Representative immunoblots of input proteins (left) and after Flag pull-down of ectopically expressed DARS2 (right) are shown. Source data are provided as a Source Data file.

## FBXO24 targets DARS2 for ubiquitylation and degradation

We next assessed DARS2 as a substrate for FBXO24-mediated cellular disposal. Using a CRISPR generated *FBXO24* KO BEAS-2B cell line, compared to a control cell line, higher levels of DARS2, and TOM20 were selectively observed and FBXO24 protein was sufficiently reduced (Fig. 4a, Supplementary Fig. 4a). Endogenous DARS2 protein exhibited a ~ 2 h half-life (t ½) and levels accumulated when cells were treated with bafilomycin A1 suggesting disposal within the lysosome (Fig. 4b, c). Further DARS2 levels accumulated maximally when treated with the pan-ubiquitin inhibitor MLN-7243 for 30 min or 1 h showing ubiquitin dependent regulation of DARS2 levels (Fig. 4d). DARS2 t ½ was extended in BEAS-2B cells stably integrated with CRISPR sgRNA against *FBXO24* compared to control cells (Fig. 4e). DARS steady-state mass decreased in a time and dose dependent manner in cells stably integrated with doxycycline inducible *FBXO24* (Fig. 4f, g). Ectopically expressed *FBXO24* was sufficient to increase DARS2 polyubiquitylation in co-immunoprecipitation studies, and FBXO24 ubiquitylated recombinant DARS2 in vitro (Fig. 4h, i). To assess the pattern of polyubiquitylation chain formation, cells were co-transfected with HA-*ubiquitin* plasmids harboring mutations at specific lysines with Flag-*DARS2* plasmid and with or without ecto-pically expressed V5-*FBXO24* and cells processed for pull-downs. With the exception of K48 polyubiquitylation, DARS2 was observed to display variable levels of polyubiquitylation linkages (Fig. 4j, Supplementary Fig. 4b).

## Molecular signatures within DARS2 modulate its lifespan

The motifs involved in conferring DARS2 protein stability have yet to be adequately described. In assessing stability of NH$_2$ terminal or carboxyl-truncated deletional constructs, DARS2 stability decreased markedly after deletion of 37 aa between P576 and D645 (Del 3) from the carboxyl-terminus or removal of the mitochondrial targeting sequence (MTS) (Fig. 5a–c). Interestingly, an extended half-life (t ½ > 8 h) of ectopically expressed *DARS2* was observed but this was reduced to t ½ = 1.17 h upon truncation in this stretch of aa suggesting loss of a stabilizing motif(s) in this region. To identify ubiquitination acceptor sites, Flag-tagged DARS2 was immunoprecipitated and analyzed via mass spectrometry (MS) using a peptide antibody-based affinity approach to identify the Lys-ε-Gly-Gly (diGLY) remnant (Fig. 5d). MS identified eight putative DARS2 acceptor sites among replicate studies, five of which had reliable spectra (Fig. 5e). In mutational analysis, where Lys was substituted with Arg (K→R), expressed mutant plasmids *DARS2*-K[153R], K[558R] and K[640R] were found to have decreased basal levels of polyubiquitylation, but a *DARS2*-K[368R] plasmid when expressed in cells showed increased polyubiquitylation (Fig. 5f). Further, DARS2-K[368R] exhibited a much more rapid decay than DARS2-WT or a stabilized double mutant, DARS2-K[558/640R] (Fig. 5g, h). Co-transfection of *FBXO24* with either *DARS2*-K[153R], K[558R] or K[640R] showed low levels of polyubiquitylation whereas a DARS2-K[368R] variant displayed high-levels of polyubiquitylation (Fig. 5i). Because lysine acetylation has been shown to protect substrates from degradation[21], we hypothesized that K[368] may act as an acceptor site for acetylation that

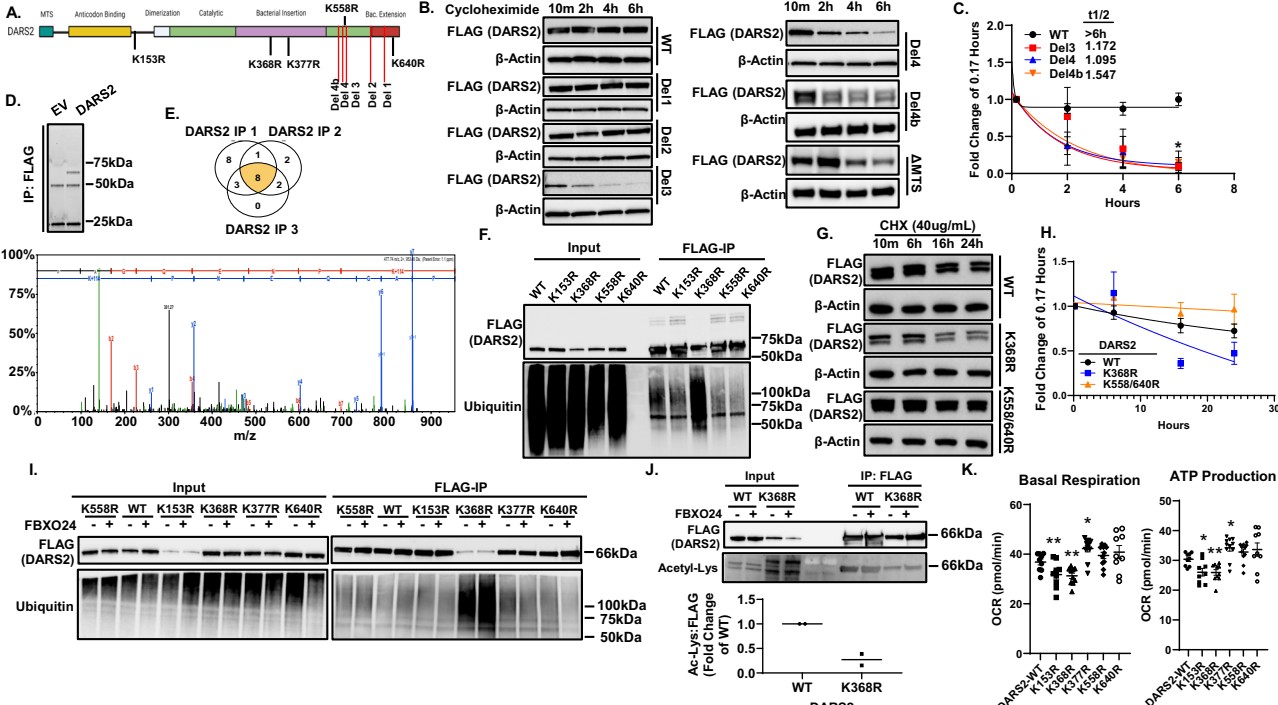

**Fig. 5 | Molecular signatures impact DARS2 stability. A** Structural map showing regions of DARS2 essential to its stability with indicated sequential, cumulative deletions of the carboxyl-terminus and deletion of its NH$_2$-terminal mitochondrial targeting sequence (MTS) and candidate point mutants at lysine residues. The span of residues from deletion mutants is shown in the Methods. **B** Transient over-expression of deletion mutants in HEK293T cells and treatment with CHX (40μg/mL) for 10 min, up to 6 h to assess construct stability with representative immunoblots and quantification of t ½ by densitometry **C** (n = 3). **D** IP of ectopically expressed empty vector (*EV*) or *DARS2* in HEK293T cells shows a - 66 kDa band processed for Mass Spectrometry (MS) (n = 3 biologically independent samples). **E** Graphical representation of ubiquitylated lysine residues shared in separate experiments from DARS2 IP replicates showing one of the validated MS spectra below. **F** Ubiquitylated DARS2-WT and K→R DARS2 variants after transient over-expression of plasmids in HEK293T cells and IP with Flag-antibody conjugated magnetic beads. Shown is a representative immunoblot (n = 3). **G** Transient over-expression of K→R *DARS2* mutants in HEK293T cells and treatment with CHX for 10 min up to 24 h is shown to assess t ½. Shown are representative immunoblots and densitometric analysis of decay in **H** (n = 3). **I** Immunoblot of IP with anti-Flag beads to measure the effect of *FBXO24* plasmid transfection (1 μg) on poly-ubiquitylation levels of DARS K→R mutants or DARS2 WT (n = 3). **J** Acetylation levels of Flag-*DARS2*-WT or *DARS2*^K368R plasmid co-expressed in HEK293 cells with *FBXO24* (1 μg) or a control plasmid. Cells were processed for Flag IP and immunoblotted with acetyl-Lys antibody and a representative immunoblot shown with and quantification below (n = 2). **K** Mitochondrial function was assayed using a Seahorse XFe96 analyzer to determine behavior of *DARS2* mutant plasmid overexpression (n = 9 biologically independent samples per group). Basal Respiration (K153R p = 0.0095, K368R p = 0.0019, K377R p = 0.0081), ATP Production (K153R p = 0.0298, K368R p = 0.0027, K377R p = 0.0133) *P*-values determined by individual two-tailed Student's t-tests. **C, H** and **K** data are presented as mean values ± SEM. Source data are provided as a Source Data file.

serves to stabilize DARS2. Consistent with this hypothesis, co-immunoprecipitated DARS2-K^368R displayed decreased acetylation compared to WT DARS2 (Fig. 5j). Cellular expression of an acetylation mimic, DARS2-K^368Q displayed a t ½ comparable to WT DARS2 (Supplementary Fig. 4c). *DARS2*-K^368R overexpression in BEAS-2B cells impaired basal respiration and ATP production compared to DARS2-WT (Fig. 5k). Collectively, the data suggest that DARS2 acetylation, together with an NH$_2$- terminal MTS or carboxyl-terminal stabilizing motifs, protect the enzyme from SCF^FBXO24-mediated cellular degradation.

## Extracellular DARS2 exhibits immunostimulatory activity

DARS2 appears essential to cell viability in neurons and cardiomyocytes[6,22,23] and has been suggested to modulate macrophage-dependent wound healing[16,22]. However, the biological relevance of DARS2 in experimental pneumonia is unknown. Knockdown of *DARS2*, but not related mt-aaRS in BEAS-2B cells significantly attenuated the release of the innate immune cytokines IL-1β, IL-6, and TNFα in response to infection with PA103 (Fig. 6a–c and Supplementary Fig. 5a–c). *DARS2* knockdown in BEAS-2B cells also significantly impaired wound healing and cell cycle progression (Fig. 6d–f and Supplementary Fig. 5d).

In the process of assessing DARS2 trafficking, we detected the protein within the culture medium, and further analysis detected the aspartyl-tRNA synthetase within secreted exosomes in response to LPS or Pam3CSK4 stimulation (Fig. 6g). The secreted DARS2 appears to migrate faster on SDS-PAGE gels suggesting release of a potential cleavage fragment during or prior to secretion (Fig. 6g, Supplementary Fig. 5). THP-1 or BEAS-2Bs cells treated with Pam3CSK and LPS showed dose and time dependent secretion of DARS2 into the supernatant (Supplementary Fig. 5e, f). To examine the effect of secreted DARS2 on cytokine release, primary human CD14+ cells were differentiated into macrophages using macrophage colony stimulating factor (M-CSF) and treated with increasing concentrations of recombinant human DARS2 (rhDARS2) or lipofectamine (vehicle) (Fig. 6h–j). These cells displayed a significant increase in IL-1β release from cells treated with rhDARS2 at 24 h post-treatment compared to vehicle controls (Fig. 6h) and dose dependent increases in IL-6 and TNFα release at 6 h and 24 h post treatment (Fig. 6i, j). To investigate these results further, vehicle, rhDARS2 or AARS2 packaged in lipid vesicles were delivered intra-tracheally (i.t.) to mice. Mice receiving rhDARS2 had significantly increased immune cell infiltration compared to vehicle controls at 4 h and 24 h post-treatment (Fig. 7a); polymorphonuclear (PMN) (Fig. 7b) and mononuclear (Fig. 7c) cells increased within BAL in contrast to

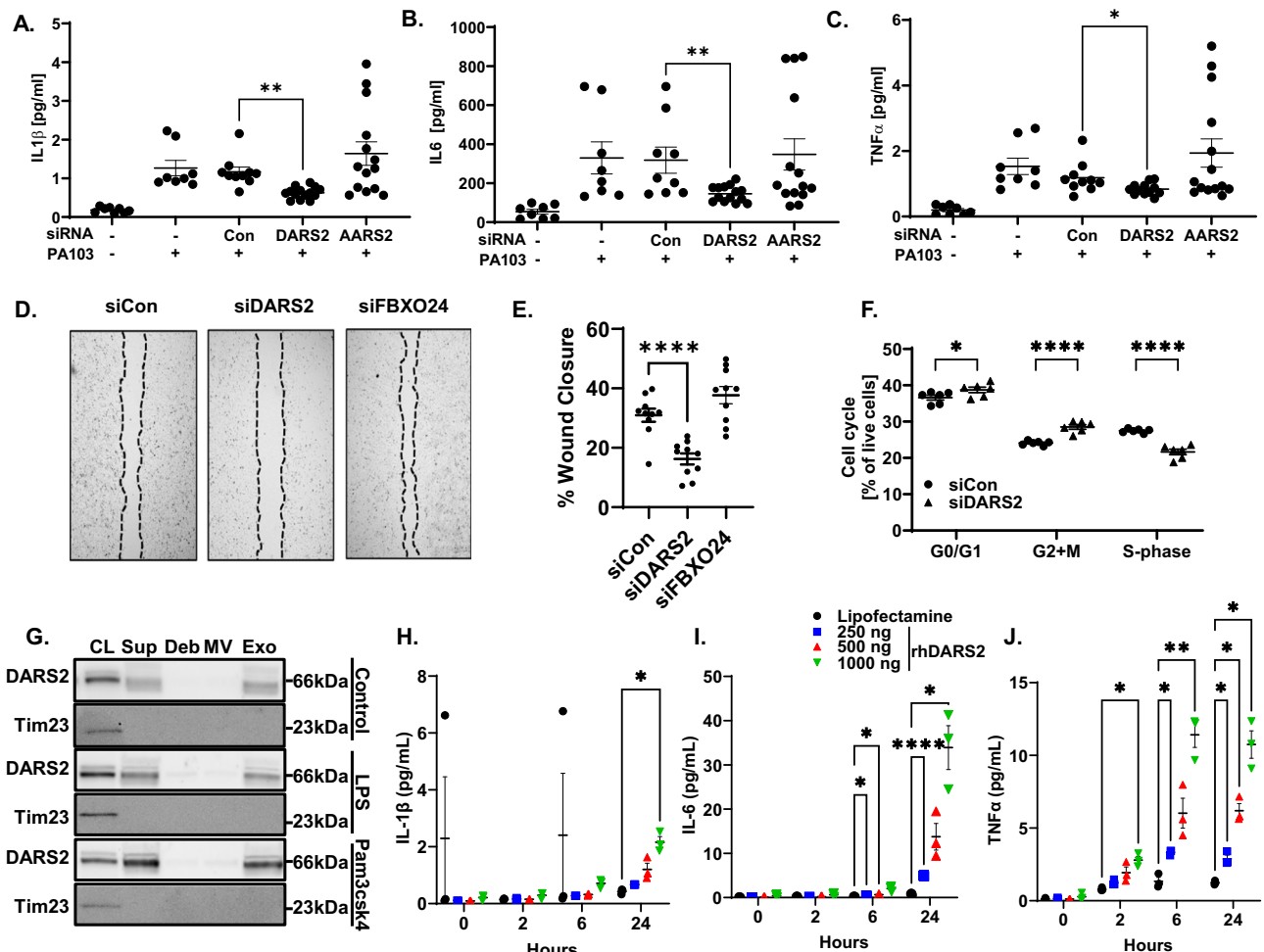

**Fig. 6 | DARS2 is a secreted immunostimulatory protein. A–C** BEAS-2B were transfected with control RNA, *DARS2* siRNA, or siRNA targeting AARS2 for 72 h, then infected with PA103 (MOI 10 for 6 h); supernatant was collected and assayed for cytokine levels (*n* = 8-14 biologically independent samples/group across *n* = 3 experiments, IL-1β p = 0.0096, IL-6 p = 0.0049, TNFα p = 0.028). **D** Representative images and quantification of BEAS-2B cells treated with 10 nM control RNA, *DARS2* siRNA or *FBXO24* siRNA and wound healing tested by a scratch assay and results quantitated **E.** (*p* < 0.0001), (*n* = 10 biologically independent samples from 2 repeats). **F** Cell cycle analysis via BrdU incorporation and flow cytometry of BEAS-2B treated with 10 nM control RNA or *DARS2* siRNA. (G0/G1 p = 0.05, G2 + M p < 0.0001, S *p* < 0.0001, by student's t-test) (*n* = 6 biologically independent samples from 2 repeats). **G** Shown are effects of LPS (500 ng/ml) or Pam3CSK4 (200 ng/ ml) on DARS2 protein within cell lysates (CL) or secreted into supernatants (Sup), debris (Deb), macro vesicles (MV) and exosomes (Exo) by BEAS-2B cells (*n* = 3). **H–J** Shown are IL-1β, (*p* = 0.0187), **H**, IL-6, (6 h: 250 ng p = 0.0208, 500 ng p = 0.0161, 24 h: 250 ng p < 0.0001, 1000 ng p = 0.0413), **I**, and **J** TNFα, (2 h 1000 ng p = 0.0237, 6 h 250 ng p = 0.0204, 1000 ng p = 0.0084, 24 h 250 ng p = 0.0351, 500 ng p = 0.0161, 1000 ng p = 0.0179), levels in the supernatants of differentiated CD14+ cell cultures treated with increasing amounts of recombinant human DARS2. *n* = 3 biologically independent samples. **A–C, E, F, H–J** data are presented as mean values ± SEM. **A–C, E, F** *p*-values derived from unpaired Student's t-test two-tailed between noted groups. **H–J** Two-way ANOVA with Dunnett's multiple comparisons test. Source data are provided as a Source Data file.

vehicle and AARS2 groups at 24 h after instillation coupled with a significant increase in IL-6 (Supplementary Fig. 5g). Mice receiving rhDARS2 displayed significant increases in protein levels of IL-6, IP-10, KC, MIP2, IFNγ, IL-1β, IL-5 and MCP-1 in their BAL compared to vehicle controls at 4 h (Fig. 7d). rhDARS2 also significantly increased numbers of lung tissue inflammatory cells and lung injury compared to control 24 h post-instillation (Fig. 7e, f). As an alternative approach, rhDARS2 was injected i.p. into mice within lipid vesicles or vehicle. Mice receiving rhDARS2 showed a significant increase in BAL cell numbers and in peritoneal fluid (PF) compared to vehicle (Fig. 7g, h) To determine clinical relevance of these observations, we examined the levels of DARS2 in the plasma of noncritically-ill (*n* = 10) and critically ill patients (*n* = 40) in the intensive care unit who were positive for viral or bacterial infections (*n* = 50, Supplementary Table 1). The infected patients displayed significant increases in DARS2 levels compared to controls at 1, 7, and 21d post admission (Fig. 7i). Similarly, in a cohort of patients that were culture positive for *P. aeruginosa*, DARS2 was also

significantly elevated in the plasma 7d post admission (Fig. 7j). Overall, mortality was significantly higher in the critically ill patients (Supplementary Table 1). Thus, DARS2 is secreted in plasma during humans with bacterial pneumonia, perhaps as a compensatory response, where it elicits host protective immunostimulatory effects.

## Design and testing of a FBXO24 inhibitor

To preserve DARS2 function, we embarked on a virtual drug screen to identify a FBXO24 inhibitor. We first predicted the FBXO24 crystal structure using Alphafold. FBXO24 contains a major C-terminal β-propeller domain that might be important for substrate targeting (Fig. 8a). We hypothesized that a small molecule inhibitor engaging the FBXO24 β-propeller domain would disrupt Fbxo24:DARS2 interaction, thus preserving DARS2 levels in cells. Using molecular docking analysis and score-ranking operations, we assessed potential ligands that might fit the FBXO24 domain cavity. Through the LibDock program from Discovery Studio 4.1, we were able to virtually screen 100,000

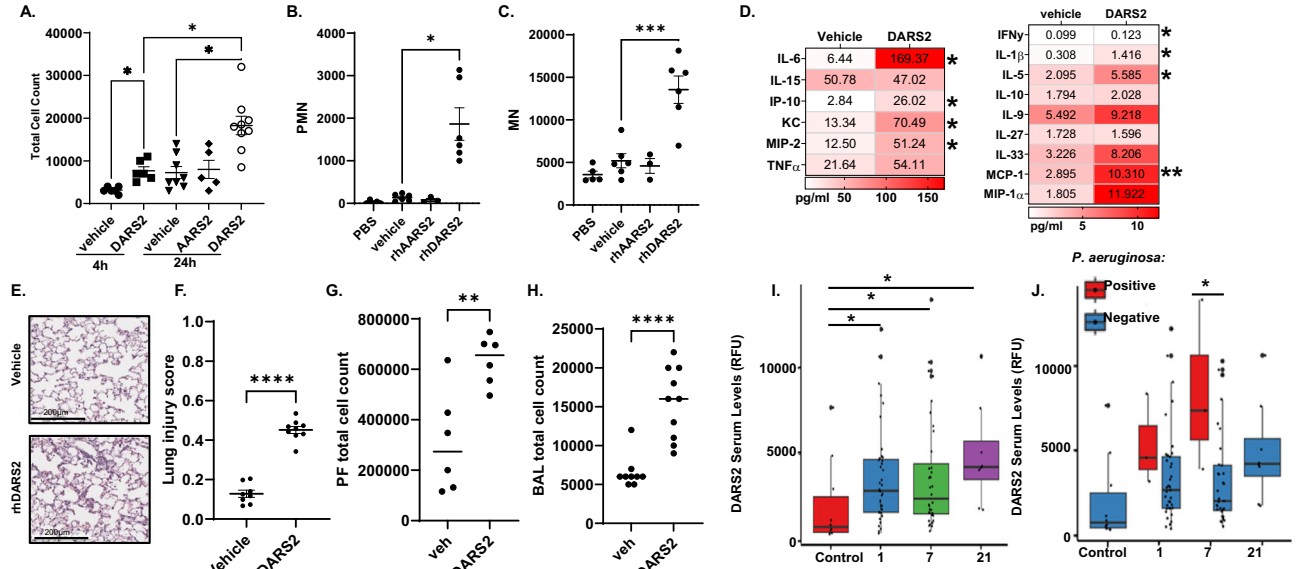

**Fig. 7 | Extracelluar DARS2 stimulates innate immune responses in vivo. A** Total cell counts in BAL of mice analyzed either 4 h or 24 h after i.t. treatment with vehicle (protein transfection reagent), recombinant human DARS2 (5 μg), or recombinant AARS2 (5 μg) packaged in lipid vesicles (*n* = 5–9/group) (4 h Veh-DARS2 *p* = 0.0402, 24 h Veh-DARS2 p = 0.0122, DARS2 4 vs.24 h p = 0.0119). **B**, **C** Shown are PMNs (*p* < 0.0001) **B** and mononuclear cells (*p* = 0.0003) **C** levels in BAL of mice from the 24 h cohort. **D** BAL cytokines in mice given recombinant DARS2 (5 μg i.t.,) and analysis at 4 h post-treatment (*n* = 3-6/group), (IL-6 *p* = 0.0388, IP-10 *p* = 0.0331, KC *p* = 0.0403, MIP2 *p* = 0.0326, IFNy *p* = 0.0298, IL-1β *p* = 0.0373, IL-5 *p* = 0.0247, MCP-1 *p* = 0.0071). **E** Lung histology and **F** lung injury score (*p* < 0.0001) in mice treated with vehicle or recombinant DARS2 (5 μg i.t.) from the 24 h cohort (*n* = 8-9 animals/group). **G**, **H** Shown are peritoneal fluid (PF) (*p* = 0.0051) **G** or BAL (*p* < 0.0001) **H** cell counts of mice injected with vehicle or recombinant DARS2

(5 μg i.p.) analyzed 24 h after injection (*n* = 5-8/group). **I** Relative DARS2 protein abundance was assayed from plasma of critically ill patients at multiple time points during ICU stay (days 1, 7, 21) and control patients without critical illness (Day 1 p = 0.019, 7 *p* = 0.014, 21 *p* = 0.021). **J** Among critically ill patients, relative DARS2 protein levels increased over time compared to controls (left) with subgroup analysis demonstrating DARS2 abundance in patients with and without *P. aeruginosa* infection (*p* = 0.034), *n* = 50. **A**–**C**, **F**–**H** data are presented as mean values ± SEM. **A**–**C** *p*-values derived from Welch's ANOVA with Dunnett's T3 multiple comparisons. **D**, **F**–**H** Analyzed with unpaired Student's t-test, two-tailed. **I**, **J** Mann-Whittley two-tailed test used for analysis and data presented as IQR for minima and maxima, mean as center and whiskers 1.5xIQR range. Source data are provided as a Source Data file.

potential ligands for the FBXO24 pocket. The top fifteen score-ranking molecules were selected and further evaluated using in vitro experiments. After an initial round of screening, we identified a "hit" compound, termed BC-1293 for further testing. Shown is the structure and in silico docking model of BC-1293 bound to FBXO24 (Fig. 8b, c). In this model, Arg[138], Trp[413] and Leu[364] residues within Fbxo24 are important for interacting with inhibitors (Fig. 8d). To first assess the ability of BC-1293 to abrogate FBXO24:DARS2 binding as a measure of molecular target validation, we transfected cells with V5-*FBXO24* plasmid, isolated expressed V5-FBXO24 protein from cells onto magnetic FLAG beads prior to incubation with increasing concentrations of the compound or a control (BC-1395), followed by incubation with rhDARS. The preparations were washed, and eluted protein complexes remaining bound to the beads were processed for immunoblotting. As shown, unlike BC-1395, BC-1293 at or above 250 nmol reduced DARS2:FBXO24 binding whereas DARS1 did not interact with FBXO24 in the presence or absence of the compound (Fig. 8e). Inclusion of BC1293 in the culture medium also extended DARS2 t½ (Fig. 8f, Supplementary Fig. 6a). Interestingly FBXO24 was found to be a highly unstable protein with a t ½ < 2 h (Fig. 8f, Supplementary Fig. 6a). Indeed, the difference in t ½ of endogenous and exogenous DARS2 (Figs. 4b and 5) may be partly attributed to the short t ½ of FBXO24 where the pool of ectopically expressed synthetase exceeds degradative capacity. Further, because FBXO24 was short-lived and induced after a bacterial stimulus (Fig. 1), additional experiments testing BC-1293 in cells were conducted with the inclusion of Pam3CSK4 as a means to increase endogenous levels of the F-box protein for drug targeting to capture its cargo. Under these experimental conditions, BC1293 significantly increased DARS2 protein mass in wild-type, but

not *FBXO24* silenced BEAS2B cells (Fig. 8g, h). These results indicate an FBXO24-dependent effect and support target validation. BC-1293 potentiated Pam3CSK4 induced secretion of both endogenous and ectopically expressed Flag-*DARS2* from cells (Fig. 8i). BC-1293 significantly increased secretion of IL-6, TNFα, and IL-1β from BEAS-2B, CD14 +, and THP-1 cells respectively, compared to vehicle or Pam3CSK4 alone (Fig. 8j-l). BC-1293 also triggered release of TNFα and IL-1β from BEAS2-2B and CD14+ cells, respectively (Supplementary Fig.6 b,c). Mice treated with BC1293 had significant increases in BAL total cell numbers and mononuclear cells without changes in PMN cells (Fig. 8m–o). BC1293 significantly increased IL-1β, IL-9, MIP-2, and TNF-α protein concentrations in BAL of mice compared to vehicle controls (Fig. 8p). Thus, the F-box inhibitor was sufficient to enhance innate immune responses in vivo. Finally, to further validate FBXO24 targeting of BC-1293 to induce cytokine secretion, we utilized precision cut lung slice (PCLS) cultures from *Fbxo24* WT or *Fbxo24* heterozygous (Fbxo24[+/-]) mice. *Fbxo24*-WT PCLS co-treated with Pam3CSK4 (5 μg/mL) with increasing concentrations of BC-1293 (0, 1, 10 and 20 μM) displayed a significant stimulatory effect of BC-1293 on IL-6 secretion 24 h post treatment (Fig. 8q). This effect was not observed in PCLS derived from *Fbxo24*[+/-] mice (Fig. 8r) demonstrating that BC-1293 immunostimulatory activity is FBXO24 dependent.

## Discussion

An unmet need in bacterial pneumonia is the identification of unique pathways that might underlie impaired immunity observed in patients[4]. The contributions here suggest that (i) DARS2 is a secreted, acetylated biomolecule with immunostimulatory properties found in the plasma of people with pneumonia, (ii) the cellular abundance of

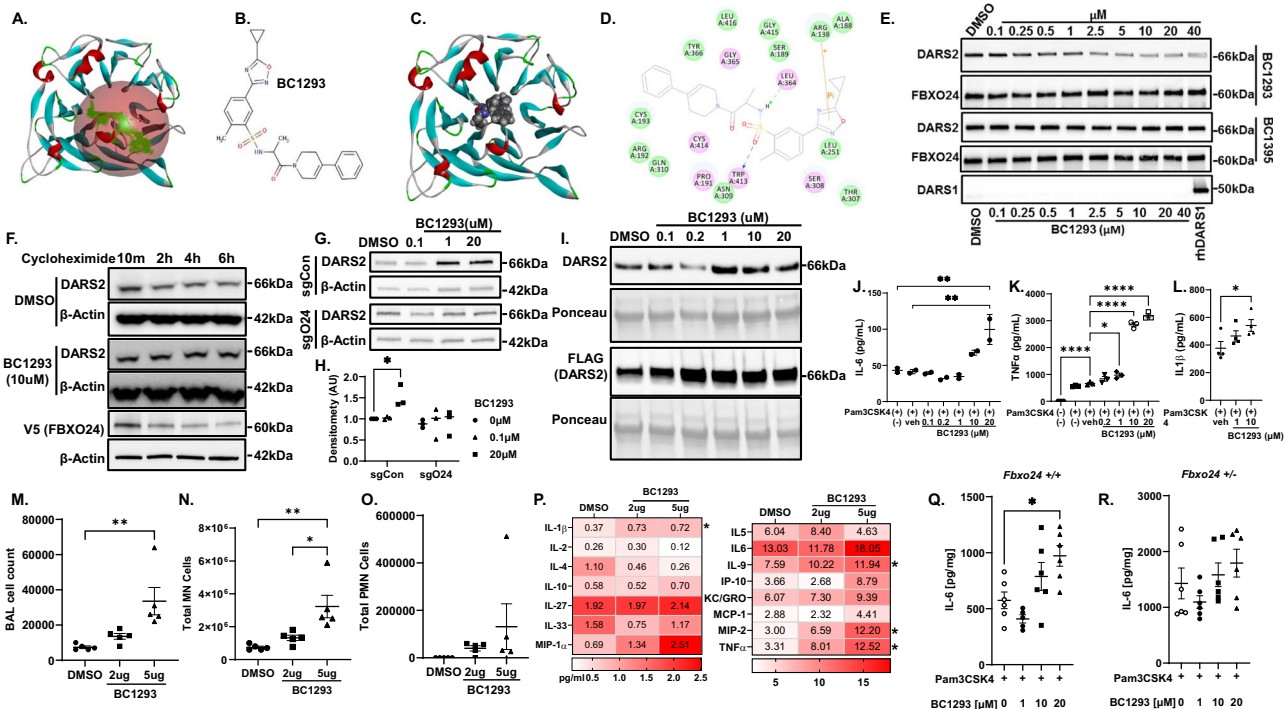

**Fig. 8 | A FBXO24 inhibitor stabilizes DARS2 and triggers immune responses.**
**A** Predicted crystal structure of FBXO24 beta-propeller domain. **B** BC-1293 chemical structure. **C** In silico docking model of potential ligand binding to FBXO24 β-propeller domain. **D** 2D diagram showing key interacting residues of FBXO24 with ligand. **E** A representative immunoblot of recombinant DARS2 binding with exogenous FBXO24 from IP from HEK293 cells incubated with increasing concentrations of BC-1293 or a specificity control BC-1395 ($n = 3$). Below, DARS1 was not detected in IP, with rhDAR1 as a loading control. **F** DARS2 and t ½ in cells treated with DMSO or BC-1293 ($n = 3$). **G, H** DARS2 levels in control cells (sgCon) or *FBXO24* depleted (sgO24) BEAS-2B cells treated with Pam3CSK4 and BC1293 (0.1-20 μM) or vehicle control (DMSO) with PamCSK4 ($n = 3$) **H** Densitometric analysis from **G** ($p = 0.038$). **I** Endogenous (top blot) or ectopically expressed Flag-*DARS2* secretion (lower blot) from BEAS-2B cells or HEK293T cells, respectively, into supernatants with inclusion of Pam3CSK4 and BC-1293. Control cells were exposed to DMSO with Pam3CSK4 ($n = 3$) **J–L** IL-6, TNFα, or IL-1β secretion from BEAS-2B (top

$p = 0.0029$, lower $p = 0.0024$,) **J** CD14+ (from top $p < 0.0001$, $p < 0.0001$, $p = 0.0117$, $p < 0.0001$) **K**, or THP-1 ($p = 0.0438$) **L** cells, respectively, in cells treated with BC-1293 or vehicle (veh) in the absence or presence of Pam3CSK4 (1 μg/ml)($n = 3$). **M–P** Mice treated with DMSO control or BC-1293 (2 μg or 5 μg i.t.) for 24 h ($n = 5$/group) were analyzed for BAL total cell counts ($p = 0.0016$) **M**, mononuclear (top $p = 0.0027$, lower $p = 0.0159$) **N**, polymorphonuclear cells **O** and **P** cytokines ($n = 5$/group) (IL-1β, DMSO-2μg, $p = 0.0261$; IL-9, DMSO-5μg, $p = 0.0261$; MIP-2 DMSO-2μg, $p = 0.0469$; TNFα, DMSO-2 μg, $p = 0.0422$). **Q, R** IL-6 in culture medium from precision cut lung slices from *Fbxo24* WT ($p = 0.0145$) **Q** or *Fbxo24* heterozygous (*Fbxo24*$^{+/-}$) mice **R** co-treated with Pam3CSK4 (5 μg) and increasing doses of BC1293 (24 h). $n = 5–6$ lung slices/group. **H**, **J–O** and **Q, R** data are presented as mean values ± SEM. **J–O** and **Q, R** Ordinary one-way ANOVA with Tukey's **J, M, N** or Dunnett's **K, L** multiple comparisons, **P** Ordinary Student's t-test two-way. **G** Two-way ANOVA with Tukey's multiple comparisons test. Source data are provided as a Source Data file.

this enzyme is reduced, in part, by the SCF$^{FBXO24}$ machinery, and (iii) an FBXO24 small molecule antagonist restores DARS2 levels and displays immunostimulatory activity. Through DARS2 disposal in cells we uncover a previously undescribed immunoregulatory function of a ubiquitin E3 ligase subunit that can be exploited to develop non-antibiotic therapeutics for bacterial infections.

The behavior of DARS2 as a secreted protein with immunostimulatory activity may reflect an intrinsic defense mechanism in response to systemic or localized bacterial infection. Hence, we observed significantly increased levels of circulating DARS2 for weeks in a cohort of patients with bacterial culture-positive infections. DARS2 acts as a modulator of neuro-inflammation and macrophage wound healing[16,22], but its direct pro-inflammatory action after secretion has not been recognized. Extracellular DARS2 may represent a more complex innate host protective role that modulates systemic, paracrine, or even autocrine signaling to elicit cellular repair or anti-microbial activity. The release of this synthetase from epithelia was selective as related mt-aaRS did not elicit similar effects. Further, the ability of DARS2 to trigger cytokine release from primary human CD14+ macrophages attest to its paracrine behavior.

To our knowledge, DARS2 is the first mitochondrial tRNA-synthetase to modulate host immunity. Interestingly, the activity of mitochondrial DARS2 resembles that of a distinct family of

cytoplasmic tRNA-synthetases that act in a similar manner. For example, the cytoplasmic tryptophan-tRNA synthetase (WRS) is secreted and primes macrophages leading to increased phagocytosis in both viral and bacterial infections[24–26]. WRS is secreted as both a full length and "mini" peptide with the former being released in response to infection. Our observed slower and faster migrating bands from immunoblots from supernatant proteins from cells may represent similar stimulus-dependent secretion of cleaved DARS2 fragments that merits further investigation. Similarly, lysyl-tRNA synthetase is secreted in response to TNFα[27] and Shiga toxin[28] to bind to macrophages inducing proinflammatory cytokine and chemokine production in a feed forward mechanism. Cytoplasmic glutamine-proline tRNA synthetase acts bi-directionally in viral infections to complex with mitochondrial antiviral-signaling protein to protect it from degradation thereby enhancing viral clearance[29]. The latter synthetase also is part of the gamma activated inhibitor of translation complex to suppress pro-inflammatory gene translation[30]. Our findings reveal that a mitochondrially-derived synthetase, perhaps because of its evolutionary conserved sequences with bacteria[31], may harbor moonlighting activities both in mitochondrial protein translation and as a potent immunostimulatory mediator resembling damage associated molecular patterns (DAMP) is intriguing. In this regard, DARS2 has a high degree of homology with its bacterial ancestors; its closest relative

aspartyl-tRNA synthetase from *E. coli* has a similar 3D structure and 43 percent homology of primary sequences[32] with DARS2 that may contribute to its potential as a DAMP. Alternatively, DARS2 may be supporting immune responses through mitochondrial biogenesis via protein synthesis as these functions are required for a competent immune response such as B-lymphocyte mobilization[33–36].

FBXO24 essentially acts as an immunosuppressive protein through DARS2 binding, ubiquitylation, and cellular degradation which demonstrates a unique ability of an E3 ligase component targeting a mitochondrial protein to regulate immunity. E3 ubiquitin ligases, for example, suppress innate immune or adaptive responses by negatively regulating T-cell activation[37] or the inflammasome[38]. Both genetic ablation or chemical inhibition of *Fbxo24* enhanced cytokine secretion and increased pulmonary cellular infiltration with varying effects on causing tissue injury depending on the experimental context. While FBXO24 immunosuppressive activity may be detrimental to pathogen clearance at early stages of infection, it may also protect against collateral tissue damage and aid in the resolution of inflammation in pneumonia similar to the purported functions of lipid mediators and interleukin 10[39,40]. Interestingly, while FBXO24 was observed to be induced in transplant-rejected lung samples positive for bacterial infection, we also observed increased plasma levels of DARS2 in infected patients. These results suggest that potentially reduced DARS2 levels after FBXO24-mediated degradation in the lung might trigger a compensatory response in circulation to enhance bacterial clearance. Further studies examining the kinetics of FBXO24 expression and DARS2 abundance in experimental pneumonia and in humans will elucidate the biological association for these proteins in host immunity.

DARS2 degradation by the SCF$^{FBXO24}$ ligase appears complex with several molecular signatures within both the NH$_2$-terminus and carboxyl-terminus. Disruption of the MTS may impair mitochondrial trafficking of DARS2 that is also essential for its stability. A putative stretch of 37 stabilizing residues within the tRNA synthetase carboxyl-terminus also impacts DARS2 lifespan. Although speculative, this region is enriched with hydrophobic and basic residues outside the bacterial extension domain that when deleted, may be important in membrane tethering to mask the tRNA synthetase from FBXO24 targeting. Importantly, within the bacterial insertion domain we identified a putative molecular site for DARS2 acetylation based on mutational analysis and testing of an acetylation mimic. The observation that DARS2 is robustly polyubiquitylated after mutation of this site with reduced acetylation underscores molecular interplay between post-translational modifications in controlling DARS2 cellular concentrations.

The emergence of multi-drug resistant bacterial pathogens remains a significant challenge in the biotechnology of drug development[41–43]. Here, through virtual screening, we identified a tool compound that was sufficient to stabilize DARS2 and increase steady state levels in cells. This effect of the compound appears dependent on its interaction with FBXO24 as epithelial cells lacking FBXO24 did not demonstrate BC-1293 induction of DARS2. Further, mouse lung slices isolated from wild type mice stimulated with Pam3CSK4 and then treated ex vivo with BC-1293 stimulated IL-6 release, but not in lung slices from Fbxo24$^{+/-}$ mice. In addition, BC-1293 added to protein binding reactions selectively impaired DARS2:FBXO24 interaction. Together, the results strongly support target validation of the compound. However, confirmation of direct molecular interaction of BC-1293 with FBXO24 to assess binding affinity requires highly purified and properly folded FBXO24 that is needed for isothermal calorimetry or surface plasmon resonance studies. We have had substantial challenges with scale-up of E3 ligase components, including FBXO24 with adequate purity, yield, and conformation suitable for binding studies. Nevertheless, such studies will be important for compound characterization moving forward.

Biologically, BC-1293 was sufficient to trigger cytokine and cellular responses in vivo. BC1293 increased macrophage influx in the lung with concomitant release of macrophage inflammatory protein (MIP-2) in BAL. Interestingly, BC1293 showed less potent immunostimulatory activity in cell cultures but was sufficient to induce both cytokine and chemokine secretion and immune cell recruitment in mice without co-stimulation with Pam3CSK4. This suggests that the small molecule's more limited effects on cytokine secretion in monocultures may reflect its actions in innate signaling in a paracrine rather than autocrine manner. Future studies involving modifications within BC1293 structure to optimize its structure-activity response in innate immune signaling and target validation will be essential in generating a new genus of tRNA synthetase activators. The studies also raise the opportunity for testing utility of DARS2 recombinant peptides in immunodeficient patients with respiratory infections.

## Methods

### Ethics statement
Mouse studies were conducted under the supervision of the University Laboratory Animal Resource. All studies were approved by the Ohio State University Institutional Animal Care and Use Committee under protocol number 2019A00000019-R1. Human lung tissue was obtained from failed donors under Institutional Research Ethics Board approval at The Ohio State University. We investigated existing banked biospecimens from 40 patients with acute respiratory failure who were admitted to the medical intensive care unit (MICU) at the Ohio State University Wexner Medical Center and The James Cancer Hospital between May 2020 and December of 2021. After obtaining informed consent from patients or their legally authorized representatives, biospecimens and clinical data was collected and stored through the Ohio State University ICU Registry (BuckICU) (IRB protocol #2020H0175). Patients were enrolled within 48 h of ICU admission. Clinical data was coded with sample ID and no uncoded data was shared with the investigators of the study.

### Cell culture and reagents
Human cell lines A549, BEAS-2B, HEK293T and THP1 cells were purchased from ATCC. Human CD14+ monocytes were purchased from Lonza. A549 and THP1 cells were cultured in RPMI (Gibco) supplemented with 10% FBS, HEPES, L-glutamine, penicillin/streptomycin, non-essential amino acids and sodium pyruvate (Gibco). BEAS-2B cells were cultured in HITES media: DMEM/F12 supplemented with 10% FBS, insulin, transferrin, hydrocortisone, β-estradiol, HEPES, L-glutamine and penicillin/streptomycin. Tet-inducible Luc and FBXO24 BEAS-2B cells were selected and cultured in HITES media with puromycin (2 μg/mL). HEK293T cells were cultured in DMEM supplemented with 10% FBS, HEPES, L-glutamine, penicillin/streptomycin. CD14+ primary cells were cultured in RPMI with 50 ng/mL Macrophage Colony Stimulating Factor for a week prior to use. Antibodies used for immunoblotting were anti-DARS2 (ProteinTech, 1:2000 dilution), anti-FBXO24 (Novus, 1:1000 dilution), anti-β-actin (1:5000 dilution, Sigma Aldrich) and the following antibodies were all purchased from Cell Signaling Technologies: TOM20 PGC1α, GAPDH, Cyclin-D1, Flag, and V5 all used at 1:1000 dilution. Other antibodies include FBXO45 (Aviva Systems Biology), FBXL2 (Aviva Systems Biology), FBXL18 (Ab-Nova), ubiquitin (Cell Signaling Technologies), IARS2 (ProteinTech), TFAM (Cell Signaling Technologies), Acetyl-Lysine 1:1000 dilution, FLAG-tag 1:1000 dilution, (both from Cell Signaling), HA-Tag 1:2000 dilution (Novus), and Tim23 1:1000 dilution (Santa Cruz Biotechnology). The recombinant human proteins DARS2, AARS2, DARS1 were from Origene and E1, E2 were purchased from Enzo Life Sciences. Plasmids include Skp-1 (GenScript, NM_170679.3), Cullin1 (AddGene, 19896), and Rbx (AddGene, 20717). An aptamer-based protein array was purchased from SomaLogic.

## Mouse Precision Cut Lung Slices (PCLS)

Age and sex matched mice were euthanized in a $CO_2$ chamber, the trachea was exposed, and a blunt needle inserted. Lungs were expanded with 1.5% of UltraPure™ Low Melting Point Agarose (Thermo Fisher, Cat#16520100) in sterile medium (DMEM, Gibco) at 50 °C. The trachea was then ligated, lungs were carefully excised, transferred to a 15 mL conical tube in cold PBS and incubated on ice for 30 min. Lungs were cut axially into 400 μM slices using a vibratome (0.30 mm/sec; Leica VT 1200), suspended in 1 mL sterile DMEM/F12 and incubated at 37 °C with 5% $CO_2$. Media was changed every hour 4 times to wash off the agarose followed by overnight incubation in DMEM/F-12 medium (Gibco) with 10% FBS (Gibco) and 1% penicillin/streptomycin solution (Gibco).

## Cell functional studies

Cell cycle progression was assayed via BrdU incorporation as a representative of total DNA volume. In brief, cells were pretreated using siRNA of interest for 48 h, followed by 4 h BrdU incorporation. Cells were then collected, fixed, permeabilized, stained and intensity was assayed by flow cytometry using the APC BrdU Flow Kit (BD Pharmingen) according to the manufacturer's instructions. Flow ctometry data was recorded with BD FACSSymphony A1 (BD Bioscience) and analyzed with FlowJo (FlowJo, LLC). Cell migration/wound healing assays were conducted using cell culture wound healing inserts (Ibidi) according to the manufacturer's instructions. Mitochondria OCR and dynamics were assayed using a Seahorse XFe96 analyzer (Agilent). A Seahorse XF Cell MitoStress Kit (Agilent) was used to conduct these experiments and were used according to protocols provided by the company. Cytokines were assayed via multiplex ELISA (Meso Scale Discovery) according to the manufacture's protocol or individual ELISAs for IL-6, IL-1β, or TNF-α (Invitrogen).

## Cell-free fractionation

Supernatants for DARS2 secretion studies were performed in cells cultured in FBS free culture media. Protein from supernatant was concentrated using Amicon Ultra-0.5 Centrifugal Filter Devices (Millipore) according to company protocol provided. Supernatant fractions were isolated via serial centrifugation. In brief 500 μL of clarified supernatant was centrifuged at 2000xg for 5 min, supernatant was saved, and the pellet (cellular debris) was washed once in cold PBS collected in protein lysis buffer. Supernatants were spun in new microcentrifuge tubes at 10,000xg for 5 min, the supernatant was collected, and pellet (MV fraction) washed and then collected. Remaining supernatants were next processed using Amicon Ultra-0.5 Centrifugal Filter Devices (Millipore) according to company protocol provided to isolate the exosome fraction.

## Protein accumulation and half-life studies

For experiments examining protein stability, cells were treated with the proteasome inhibitor MG132 [10 μM] and lysosome inhibitor Bafilomycin A1 [100 nM] for 6 h or MLN7492 [1 or 5 μM] for variable time points. Protein half-life was determined via treatment of cells with cyclohexamide [40 μg/mL] and lysates collected in a time course study.

## Plasmid and siRNA transfections

For cellular plasmid overexpression experiments, plasmid(s) were mixed with X-tremeGENE 9 (SigmaAldrich) at a ratio of 1 μg:2 μL in 300 μL OptiMEM (Gibco) per reaction. For siRNA studies, cells were transfected with a final concentration of 10 nM siRNA in a reaction mixture of 1:50 GenMute: 1x GenMute Buffer, reaction volume was dependent on well size. siRNA sequences are listed in Supplemental Data 1.

## Generation of FBXO24 deletion in cell lines and mice

We generated *Fbxo24* KO mice using CRISPR/Cas9 technology at the University of Pittsburgh. Briefly, guide RNAs were constructed and tested in blastocysts. The University of Pittsburgh Transgenic and Targeting core facility injected murine fertilized eggs with CRISPR/Cas9 RNA reagents and implanted injected embryos into pseudopregnant females. Appropriate gRNAs generated double stranded breaks resulting in a 600 bp deletion producing a non-functional allele. FBXO24 and control gRNAs are provided Supplemental Data 1. Generation of the KO mouse was confirmed by RFLP analysis and DNA sequencing. FBXO24-KO BEAS-2B cells were also generated using CRISPR technology as previously described[44]. In brief BEAS-2B cells were transfected with a plasmid encoding Cas9, and sgRNA against *FBXO24* or scrambled sgRNA and GFP. Cells were flow sorted for GFP positive cells and expanded for culture and freezing. To generate doxycycline-inducible FBXO24 cell lines, FBXO24 was cloned into a pSBtet-RP backbone containing a puromycin resistance gene under control of a RPBSA promoter. BEAS-2B cells were transfected with this plasmid and then selected with puromycin for stable integration. Luciferase control cells were generated in the same fashion.

## Bacterial preparations

Tryptic Soy Agar plates were streaked with 1–5 μL *Pseudomonas aeruginosa*, strain PA103 (ATCC) glycerol stock and grown overnight in a 37 °C bacterial incubator. An individual colony was placed into 5 mL Tryptic Soy broth and cultures were incubated at 37 °C and shaken at 250 rpm overnight. Cultures were then diluted 1:10-1:20 and cultured an additional hour and an absorbance (OD600) using Nanodrop One (Thermo Fisher Scientific) was used to determine cfu/ml. Liquid cultures were diluted for desired cfu/mL in cell culture media for in vitro or PBS for in vivo experiments. *Klebsiella pneumoniae* strain NCTC 9633 (ATCC), *E. coli* strain Seattle 1946 (ATCC 25922) *and Staphylococcus aureus* were cultured in a similar manner. *S. pneumoniae*, strain CIP 104225 (ATCC), was grown on Blood TSA plates (McKesson Medical Surgical) and cultured in Brain Heart Infusion Broth (Becton Dickenson) at 37 °C with 5% $CO_2$.

## Experimental pneumonia

All animal experiments were approved by The Ohio State University Institutional Animal Care and Utilization Committee. Mice were housed in the OSU ULAR Vivarium on a 12 h light/dark cycle at a temperature of 65–80 °F at 30–70% humidity. *Fbxo24*-KO or Wt mice from a C57BL/6 J background were given $5 \times 10^5$ cfu PA103/mouse, 3 mg/kg LPS, or PBS i.n., or for the systemic inflammation model, LPS i.p. administration (20 mg/kg) or PBS. Mice were euthanized 24 h post infection. Bronchoalveolar lavage (BAL) was collected via exposing the trachea, creating an incision, and inserting a blunted needle in the trachea. The lungs were then flushed with 1 mL PBS. BAL was then centrifuged at 1500 rpm for 10 min and supernatant transferred to a fresh tube for determination of protein concentration via Lowry assays and cytokine secretion via multiplex ELISA (Meso Scale Discovery). The remaining cell pellet was resuspended in 200 μL ACK solution (Gibco) and incubated on ice for 10 min to lyse red blood cells. To stop the reaction, 1 mL PBS with 0.1 mM EDTA was added, and the solution was centrifuged at 800 g for 10 min, supernatant was aspirated, and cells were resuspended in 400 μL PBS for differentials. Total cell count was performed with trypan blue. 200 μL of cell suspension was spun down on 2 cytology funnel slides (Fisher Scientific) per sample. Slides were processed with Giemsa staining, modified (Sigma-Aldrich) and May-Grünwald staining (Sigma-Aldrich). 100 cells/slide were counted and scored as polymorphonuclear or mononuclear cells; percent average of counts was multiplied by the total cell count and expressed as number of cells. Blood samples for multiplex ELISA were collected by puncturing the heart. Lung and live tissue were harvested for gene expression analysis.

## Bacterial loads

To determine bacterial loads mice were grouped and inoculated as described above. Mice were euthanized 24 h later, lungs were excised, collected in 1 mL sterile PBS and homogenized. Lung homogenate was diluted 1:10 8 times and 5 μL from each dilution was plated on marked sections of a TSA plate. The first dilution to display individual colonies was used to calculate bacterial. CFU/mg was calculated as: $\frac{colony\ \#\ at\ Dilution\ X * (10^{Dilution\ X}) * 200)}{mg\ of\ lung\ tissue}$

## Lung function studies

*Fbxo24* KO or WT mice from a C57BL/6 J background were given $1 \times 10^5$ cfu PA103/mouse, Lung function was assayed using a FlexiVent FX2 (SciReq). In brief, mice were anesthetized with ketamine/xylazine and received a tracheostomy to install the FlexiVent breathing tube. Once no respiratory effort was detected a mouse lung function program was run to assess several parameters of lung mechanics.

## Lung inflammation analysis

Mouse lungs were fixed in 10% formalin and sent to the Histowiz for sectioning and H&E staining. A lung injury score was quantified based on assigning 0 (none), 1 (moderate) or 2 (severe) based on individual criteria (in parenthesis) to number of infiltrating neutrophils (none, 1-5, >5), number of hyaline membranes, (none, 1, >2), proteinaceous debris in alveoli (none, 1, >2) and alveolar thickening (<2x, 2-4X, >4x). Individual scores were averaged for a final lung injury score. The scoring system used is based on the Official American Society Workshop[45]. All images were quantified by a minimum of two blinded scorers and their assessments were averaged.

## In vivo administration of recombinant protein

Recombinant human DARS2 and AARS2 were prepared in protein transfection buffer (ProJect). 5μg of recombinant human DARS2 or AARS2 was given to mice either i.t. or i.p., protein transfection reagent was used as vehicle. Mice were euthanized 4 h or 24 h after administration. BAL was collected for differential analysis and multiplex ELISA for cytokine profile. Lung tissue was collected for histology. The peritoneal cavity was washed with 10 ml of PBS/ EDTA (0.1 mM) solution and total cell counts were measured.

## In vivo administration of BC-1293

BC-1293 was prepared in PBS with 5% DMSO with a final concentration of 20 ng/μL and 50 ng/μL. C57BL6 mice were assigned to three experimental groups: DMSO, 2 μg or 5 μg BC-1293 (*n* = 5/group) and received 100 μL of respective stocks i.t. Mice were euthanized 16 h post-treatment and BAL was collected for differential analysis of immune cells and multiplex ELISA for cytokine concentration.

## Immunoblotting

Cells were collected and lysed in Protein Lysis Buffer A: PBS with 0.2% SDS, 0.05% 100X-Triton and 1 Pierce Protease Inhibitor Tablets/10 mL, (Thermo Scientific, A32963) then sonicated for 20 sec at 25% on a VibraCell Sonicator (Sonics). Protein concentration was measured by Lowry Assay. Samples were diluted with Laemmli buffer and SDS-PAGE was performed with precast gradient mini- or midi- gels and transferred to nitrocellulose membranes using a Trans-Blot Turbo Transfer apparatus. All immunoblotting supplies were from Bio Rad. Densitometry was performed using ImageJ software (NIH) or Image Lab software (Bio Rad).

## Immunoprecipitation (IP)

Prior to lysis cells were treated for 6 h with the proteasome inhibitor MG132 [100 μM] and lysosome inhibitor Bafilomycin A1 [200 nM]. Cells were lysed in protein lysis Buffer A with deubiquitinase inhibitors 1,10-phenanthroline, PR-619 and *N*-ethylmaleimide. For experiments examining acetylation, cells were pretreated as described, but Protein

Lysis Buffer A was made with 100x Deacetylase Inhibitor Cocktail (ApexBio) in place of DUB inhibitors. Lysates were sonicated and protein was normalized as described above. Cells were then incubated while rotating at RT for 30 min with anti-FlagM2 Magnetic Beads (Millipore-Sigma), washed with lysis buffer, washed twice with IP Wash buffer (0.2% Triton-X100 in PBS), and then eluted in 2x Laemmli via boiling. IP of biotinylated proteins for the TurboID screen was done as previously described[46]. For post-translational modifications studies, DARS2-Flag constructs were expressed in HEK293T cells. Cells were lysed and DARS2 was subjected to IP as described above and samples processed for analysis by MS.

## Proximity dependent biotinylation assay

As before[46], mCherry *FBXO24*-Wt and a catalytically inactive *FBXO24*-LPAA variant were cloned into a Sleeping Beauty backbone conjugated with a TurboID construct. Constructs were stably integrated into A549 cells and then treated with Biotin [50 μM] for 4 h. Samples were then subjected to IP as described above. Biotinylated proteins were identified by mass spectrometry (MS) analysis conducted by the MS and Proteomics Facility at the OSU Campus Chemical Instrument Center.

## Mass spectrometry sample processing and analysis

Beads were washed with 50 mM ammonium bicarbonate three times. Then 5uL of DTT was added and the sample is incubated at 65°C for 15 min, then 5uL of iodoacetamide (15 mg/ml) added and the sample is kept in dark at room temperature for 30 min.1ug of sequencing grade-modified trypsin (Promega, Madison WI) was added followed by incubation a*t* 37°C for overnight. The reaction is quenched the next morning by adding acetic acid for acidification. Supernatant was taken out, concentrated for LC/MSMS analysis. Nano-liquid chromatography-nanospray tandem mass spectrometry (Nano-LC/MS/MS) of protein identification was performed on a Thermo Scientific orbitrap Fusion mass spectrometer equipped with an nanospray FAIMS Pro™ Sources operated in positive ion mode. Samples (6.4 μL) were separated on an easy spray nano column (PepmapTM RSLC, C18 3 μ 100 A, 75 μm X150mm Thermo Scientific) using a 2D RSLC HPLC system from Thermo Scientific. Each sample was injected into the μ-Precolumn Cartridge (Thermo Scientific) and desalted with 0.1% Formic Acid in water for 5 min. The injector port was then switched to inject and the peptides were eluted off of the trap onto the column. Mobile phase A was 0.1% Formic Acid in water and acetonitrile (with 0.1% formic acid) was used as mobile phase B. Flow rate was set at 300nL/min. mobile phase B was increased from 2% to 16% in 105 min and then increased from 16-25% in 10 min and again from 25-85% in 1 min and then kept at 95% for another 4 min before being brought back quickly to 2% in 1 min. The column was equilibrated at 2% of mobile phase B (or 98% A) for 15 min before the next sample injection. MS/MS data was acquired with a spray voltage of 1.95 KV and a capillary temperature of 305 °C is used. The scan sequence of the mass spectrometer was based on the preview mode data dependent TopSpeed™ method: the analysis was programmed for a full scan recorded between m/z 375-1500 and a MS/MS scan to generate product ion spectra to determine amino acid sequence in consecutive scans starting from the most abundant peaks in the spectrum in the next 3 seconds. Three compensation voltage (cv = -50, -65 and -80v) were used for samples acquisition. The AGC Target ion number for FT full scan was set at 4 x 105 ions, maximum ion injection time was set at 50 ms and micro scan number was set at 1. MSn was performed using HCD in ion trap mode to ensure the highest signal intensity of MSn spectra. The HCD collision energy was set at 32%. The AGC Target ion number for ion trap MSn scan was set at 3.0E4 ions, maximum ion injection time was set at 35 ms and micro scan number was set at 1. Dynamic exclusion is enabled with a repeat count of 1 within 60 s and a low mass width and high mass width of 10ppm. Data were searched using Mascot Daemon by Matrix Science version 2.7.0 (Boston, MA) via ProteomeDiscoverer (version 2.4 Thermo

Scientific,) and the database searched against the most recent Uniprot databases. A decoy database was also searched to determine the false discovery rate (FDR) and peptides were filtered according at 1% FDR. Proteins identified with at least two unique peptides were considered as reliable identification. Any modified peptides are manually checked for validation. Label free quantitation1 was performed using the spectral count approach. The normalization scheme in Scaffold adjusts the sum of the selected quantitative value for all proteins in the list within each MS sample to a common value: the average of the sums of all MS samples present in the experiment. Student-t test was performed by scaffold (Proteome Software, Portland, OR) to evaluate if the folder change for certain proteins is significant ($p < 0.05$).

## Quick coupled transcription and translation and in vitro ubiquitylation

In vitro ubiquitination studies were conducted using Ubiquitylation kits (Enzo Life Sciences) according to manufacturer's instructions. Recombinant proteins not included in the kit were produced using a TnT Quick Coupled Transcription/Translation Systems Kit (Promega) according to the protocol included.

## Protein-compound interaction assay

In brief, Flag-tagged *FBXO24* was transiently overexpressed in HEK293T cells subjected to IP. FBXO24 bound to beads suspended in PBS was then incubated with increasing concentrations of BC-1293 or a control compound, BC-1395, for 24 h at 4 °C. Recombinant DARS2 or DARS1 were then added to the bead-FBXO24-compound reactions for 4 h at RT. Following this, beads were washed thoroughly with IP wash buffer to remove any excess unbound protein. Finally, captured protein was eluted in 2x Laemmli via boiling and samples underwent immunoblot analysis.

## Plasmids

*DARS2* primary gene sequence was isolated from cDNA from HEK293T cells and cloned into a pcDNA-3.1 backbone under the control of a EF1α promoter with a Flag tag. *FBXO24* primary gene sequence was cloned into a pcDNA-3.1 backbone under the control of a EF1α promoter with a V5 tag. *FBXO24*-LPAA mutants were generated by site directed mutagenesis. *DARS2* deletion and K → R and K →Q substitution mutants were generated through site directed mutagenesis from the original pcDNA-*DARS2*-Flag vector. The carboxyl-terminal deleted residues are shown in Table 1.

Ubiquitin mutants were generated using site directed mutagenesis from a pRK5 plasmid harboring ubiquitin WT with a HA tag. All plasmids were sequence confirmed by a core at OSU and protein expression was validated in vitro.

## Confocal microscopy

For confocal microscopy cells were cultured on culture slides (Lab-Tek). Media was aspirated, cells were fixed in 10% formalin and permeabilized with 0.5% triton and blocked with 1% BSA. Cells were then stained with primary antibody for V5 (Invitrogen), Flag (Cell Signaling

**Table 1 | Amino acid sequence removed from DARS2 C-Terminal deletions**

| Deletion Mutant | Deletion | Sequence Removed |
|---|---|---|
| Del1 | 626-645 | LKPYHIRV SKPTDSKAERAH |
| Del2 | 613-645 | LMSNTP DSVPPEE |
| Del3 | 576-645 | PP HGGIALGLDR LICLVTGSPS IRD-VIAFPKS FRGHD |
| Del4 | 543-645 | S IRIHNAELQRYILATLL KEDVKMLSHL LQALDYGA |
| Del4b | 515-645 | LYTEPK KARSQHYDLVL NGNEIGGG |

Technologies) and TOM20 (Cell Signaling Technologies) at °4 C overnight, then fluorescent secondary antibody at RT for 1 h, and counterstained with DAPI. MitroTracker Deep Red (Invitrogen) was performed according to the manufacture's protocol Images were obtained using an Olympus FV3000 Spectral Confocal System microscope. Image analysis and image preparation was done with ImageJ (NIH).

## Gene expression

To analyze changes in gene expression cells were lysed and processed using the RNeasy Plus Kit (Qiagen) according to manufacturer's protocol. Mouse and human tissue were processed using a miRNeasy Mini Kit (Qiagen). Reverse transcription reactions produced cDNA with the High-Capacity RNA-to-cDNA™ (Applied Biosystems). Expression levels were quantified by RT-qPCR using the SYBR Green system with a CFX96 Real-Time System (Bio Rad). Primers are listed in a Supplementary Table 2.

## Administration of recombinant proteins

DARS2 and AARS2 were added to Project Protein transfection reagent (ProJect) or Lipofectamine (ThermoFisher) according to manufacturer's instructions. For cell-based studies, these reactions were suspended in optiMEM and added to culture.

## Molecular docking studies and compound design

The docking experiments were carried out by using software from Discovery studio 4.1. A custom library containing 100,000 diverse molecules were first used to screen potential ligands for FBXO24. Based on the docking and best-fit analysis of suitable ligands, several compounds were tested for FBXO24 inhibitory activity including BC-1293.

## Clinical Cohort Sample Collection

Biospecimens, including peripheral blood collected in sodium citrate tubes, are collected on days 1, 7, and 21 of ICU admission. Following collection, blood is centrifuged to separate plasma and cells. Plasma is stored in aliquots at -80°C. Clinical data is adjudicated by at least 2 physician-scientists certified in critical care medicine.

## Analysis of human plasma samples

We analyzed banked plasma samples from 40 distinct patients on days 1, 7, and 21 of ICU admission. Plasma protein abundance was analyzed by an aptamer-based array using the SomaScan® platform by Soma-Logic, Inc. We used a R (version 4.2.2) statistical program to analyze data. To perform multiple comparisons of numerical variables, we utilized ANOVA for normally distributed data, and the two-sample Wilcoxon test (Mann-Whitney) or Kruskal-Wallis tests for non-normally distributed data. Chi-square tests were used to compare categorical variables among the different groups.

## RNA-Seq

RNA was extracted from cells using RNeasy Plus mini kit according to manufacture protocol (Qiagen). RNA quality, mRNA library preparation (polyA enrichment), sequencing using NovoSeq P150 (6 G raw data per sample) and data quality control was done by Novogene. ROSALIND® (https://rosalind.bio/) was used to analyze the RNAseq raw data. The HyperScale architecture has been developed by ROSALIND, Inc. (San Diego, CA). Cutadapt was used to trim the reads[47] and FastQC was used to obtain quality scores[48]. STAR was used to align the reads were to the *Homo sapiens* genome build GRCh38[49]. The quantification of individual sample reads was performed using HTseq[50] and subsequent normalization was carried out using the Relative Log Expression (RLE) method implemented in the DESeq2 R package[51]. Read distribution percentages, violin plots, identity heatmaps, and sample multidimensional scaling (MDS) plots were generated as part of the

quality control (QC) process using the RSeQC software[52]. The DEseq2 package was utilized for the computation of fold changes and p-values, as well as for the implementation of optional covariate correction. The clustering of genes for the final heatmap of differentially expressed genes was performed using the PAM (Partitioning Around Medoids) approach, utilizing the fpc R library[53]. The hypergeometric distribution was employed to assess the enrichment of pathways, gene ontology, domain structure, and other ontologies. The topGO R library[54] was employed to assess local similarities and dependencies among Gene Ontology (GO) concepts for the purpose of implementing Elim pruning correction. Differential expression analysis was performed to identify genes that exhibited significant changes in expression levels. The criteria for determining differentially expressed genes (DEGs) included a fold-change threshold of 1.5 and a statistically significant adjusted P value of less than 0.05. The heatmaps were constructed using the ROSALIND software, displaying the $\log_2$ expression values that have been standardized across the heatmap. The downstream investigation of cellular function and pathway enrichment was conducted using Ingenuity Pathway investigation, an advanced bioinformatics software tool developed by Qiagen in Ann Arbor, MI (www.ingenuity.com). Cell function enrichment of DEGs was shown as number of DEGs for each cell function cluster. Pathway enrichment was shown as percent of pathway specific DEGs vs. total DEGs with the highest predicted up- or downregulation using a cut-off of absolute |2.5| for the activation Z-score. A positive Z score predicts overall activation of the pathway, and a negative Z score predicts downregulation of the pathway, based on the number of DEGs that are up- and downregulated in each pathway cluster.

## Statistical analysis

Student's t-tests were used for experiments with two groups. One-way ANOVA with multiple comparisons were used for experiments exceeding two groups with a single independent variable. Two-way ANOVAs were used for analysis of groups with multiple independent variables, and repeated measures was used when appropriate. Recommend tests for variance of standard deviations were used and associated corrections for non-parametric data were used as needed. Outliers were removed using Data with Robust regression and Outlier removal (ROUT) method with Q = 0.1% as the threshold for removal. Statistical analysis was performed in GraphPad Prism (GraphPad) unless otherwise noted.

## Reporting summary

Further information on research design is available in the Nature Portfolio Reporting Summary linked to this article.

## Data availability

Uncropped blots and data used to generate graphs are included in the source data file uploaded with this paper. Proteomic data from this study has been deposited in the ProteomeXchange database with the identifier PXD050224. RNA-seq data has been deposited in the NCBI GEO database and will be available with the accession code GSE260733. Source data are provided with this paper.

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

## Acknowledgements

We acknowledge resources from the Campus Microscopy and Imaging Facility (CMIF) and the OSU Comprehensive Cancer Center (OSUCCC) Microscopy Shared Resource (MSR), The Ohio State University. This facility is supported in part by grant P30 CA016058, National Cancer Institute, Bethesda, MD. We also acknowledge support from the OSU Mass Spectrometry and Proteomics Core funded by P30CA016058. Human lung specimens were processed by Sean Stacey through The Ohio State University Wexner Medical Center Comprehensive Trans-plant Center Human Tissue Biorepository. This work was supported by P01HL114453, R01HL097376, R01HL081784, and R01HL096376 awarded to R.K.M.

## Author contributions

B.S.J., B.B.C., and R.K.M. designed the study, analyzed results and wrote the manuscript. B.S.J., D.F., J.A.A., R.E, J.A.J., A.C., A.E., M.E. F.M.C., L.C., F.J.J., L.R. L.F., J.S.B., and J.D.L. performed experiments and analyzed data. J.S.B. directed the human study and M.E. contributed statistical analysis for the human study. B.S.J., D.F. and R.K.M. directed animal studies. M.R. J.S.B., A.R., P.R., V.K., J.S.L. and R.K.M. edited the manu-script. R.K.M. directed the study.

## Competing interests

The authors declare no competing interests.
