## [Peer Review File · Nature Communications]

Targeted degradation of extracellular mitochondrial aspartyl-tRNA synthetase modulates immune responsesREVIEWER COMMENTS

Reviewer #1 (Remarks to the Author):

In the manuscript entitled "Targeted degradation of extracellular mitochondrial aspartyl-tRNA synthetase modulates immune responses in bacterial pneumonia", Johnson and co-workers presented the critical role of extracellular DARS2 in the innate immune response to bacterial pneumonia. They provided novel data and demonstrated the activity of DARS2 is controlled by a bacterial-induced ubiquitin E3 ligase subunit, FBXO24. In line with these findings, Fbxo24 knockout mice have elevated DARS2 levels with a robust increase in lung inflammation. In addition, BC-1293 as a FBXO24 inhibitory compound exhibited pro-inflammatory effects both in vitro and in vivo. Although the topic is novel and interesting, several major concerns affect the data interpretation and conclusions of the work.

Major concerns:

1. Fig 1i and 1j, *S. aureus* and *K. pneu* infections were not able to significantly induce FBXO24. The authors stated this may be due to these two strains being gram-positive pathogens. In fact, *K. pneu* is a gram-negative pathogen. The interpretation needs to be carefully revised.
2. Fig 2b, bacterial loads were not significantly reduced in Fbxo24 KO mice. Thus, it is inaccurate to say that "Fbxo24 KO mice showed decreased bacterial loads compared to WT mice" in the manuscript. Also, the sample size mentioned in the figure legend is n= 5-6/group. But only 4 dots were shown in the panel.
3. Fig 6g, it is very interesting to see that DARS2 has smaller MW in both sup and exo groups than CL group. It would be better to further discuss the observation.
4. There are only two biological replicates shown in Fig 1j. Thus, it is not acceptable to perform statistical analysis or make a conclusion.
5. Fig 7m-7p, is quite confusing about the procedures for these in vivo experiments. It appears that mice were only treated with DMSO control or BC-1293. But, in the introduction, it was stated that "FBXO24 inhibitor compound, that when administered in experimental pneumonia, displays immunostimulatory activities". Unfortunately, the authors did not provide any details in the Methods section.
6. One big concern comes from the specificity of BC-1293. Since the Fbxo24 KO mice are available, it would be really helpful to test the effect of BC-1293 in Fbxo24 KO mice to validate its target specificity.

Minor concerns:

1. Fig 2l, it would be better to show Total cells, PMN, or Mono in separate panels to improve the readability.
2. What is HEK293293T cell line? Is it 293 or 293T cells? Please indicate the sources of cell lines used in this study for scientific rigors and reproducibility.
3. The authors used siRNA to knock down DARS2 in Fig. 6d-f. It is inaccurate to use "depletion" when describing the result.
4. More details need to be included in Materials and Methods. For instance, the protocol used for MV, EXO isolation.

Reviewer #2 (Remarks to the Author):

In this manuscript, the authors discovered that the E3 ligase component, FBXO24, is an immunoregulatory protein via the cellular disposal of DARS2, a previously unrecognized secreted protein in human plasma with cytoprotective and immune properties. Additionally, the authors identified an FBXO24 inhibitor compound, that when administered in experimental pneumonia, displays immunostimulatory activities. This is an interesting manuscript, but I still have the following questions.

1. In Figure 1f and 1g, the data source is not clearly indicated, making it unclear whether the results in these two figures are derived from A549 cells or BEAS-2B cells. Similar issues have been found in other parts of the manuscript, which requires a careful re-examination of the original

draft, and the experimental methods in the manuscript need to be described in more detail.

2. In the description of Figure 1d-h, "...induction by PA103 infection in A549 and BEAS-2B cells after 24 h culture. " Curiously, can viable A549 and BEAS-2B cells be harvested after 24 h of PA103 treatment? In addition, BEAS-2B cells were infected with KP and SA for only 6 h, why not the same time as the previous PA103 infection?

3. In experiments to assess the pattern of polyubiquitylation chain formation, especially for the detection of K48 polyubiquitylation, did the cells use MG-132 treatment? Moreover, as a control experiment, FBXO24 can be added to the experiment in Figure 4i to determine which polyubiquitylation linkage of DARS2 is increased by FBXO24.

4. Figure 4 indicates that FBXO24 targets DARS2 for ubiquitylation and degradation, but there seems to be no direct evidence from the experimental results to suggest that DARS2 is degraded through the ubiquitination pathway.

5. The authors mention in Figure 1" Human transplant rejected lung tissue testing positive in bacterial cultures had a significant increase in FBXO24 mRNA and protein levels compared to uninfected tissue...", and in Figure 6" The critically-ill infected patients displayed significant increases in DARS2 levels compared to controls ...". "However, the author's other experimental data show that FBXO24 leads to the degradation of DARS2. Are there plausible explanations for these seemingly contradictory results?

6. In Figure 7e, BC-1293 at or above 250 nmol reduced DARS2:FBXO24 binding. Considering the degradation effect of FBXO24 on DARS2, observe whether the content of DARS2 will increase after BC-1293 reduces DARS2:FBXO24 binding. Thus, experiments can be supplemented: changes in DARS2 content in the presence or absence of BC-1293 in both infected and uninfected conditions.

Reviewer #3 (Remarks to the Author):

In this study, Johnson et al. revealed that the E3 ligase component, FBXO24, is an immunoregulatory protein via the cellular disposal of DARS2, a previously unrecognized secreted protein in human plasma with cytoprotective and immune properties. The DARS2-FBXO24 interaction mechanism may provide opportunities to modulate host immune responses through small molecule compounds. Overall, this work is a meaningful study with potential for publication. However, there are some key issues that need to be resolved before publication:

(1) Many conclusions in the article lack convincing evidence. The following are just some examples:

(a) Figure 1 includes data from only two species, which may not be sufficient to conclusively support the following assertion: Increased FBXO24 protein levels are restricted to gram-negative pathogens. Additional data are required.

(b) In terms of data significance in Figure 2, Figure 2B lacks significance, thus hampering the derivation of relevant conclusions. Additionally, there are only two values in the KO group in Figure 2E, 2F, and 2G, precluding meaningful significance calculations. The significance markers on heat maps are confusing, such as Figure 2J, some indicators that appear to be significantly different are not actually marked with significance.

(c) In Figure 6P, the concentration values of various inflammatory factors are too small. The difference between the two groups may be caused by experimental errors. It is recommended to perform additional experimental tests for confirmation.

(2) It is appreciated that the authors identified a tool compound, FBXO24 inhibitor BC-1293, through virtual screening. The authors confirmed the efficacy of BC-1293 through various cellular experiments and in vivo activity experiments in mice. However, how did the authors confirm that BC-1293 exerts in vitro and in vivo activity by targeting FBXO24? How is the dosage concentration of BC-1293 determined? Are there any data that can provide information on the binding affinity between BC-1293 and FBXO24?

(3) The concentration units in the article should be standardized, for example, ug/uM should be corrected to µg/µM.

REVIEWER COMMENTS

We have seriously considered all of the reviewers' comments and have made numerous revisions in the text and also executed several new experiments (new Fig. 1i, Fig. 4d, Fig. 7g,h,q,r, Fig S1d/e, Fig S4a,b, and additional "ns" to Fig 1i/j and Fig 2l-n) to substantively address their concerns.

Reviewer #1 (Remarks to the Author):

In the manuscript entitled "Targeted degradation of extracellular mitochondrial aspartyl-tRNA synthetase modulates immune responses in bacterial pneumonia", Johnson and co-workers presented the critical role of extracellular DARS2 in the innate immune response to bacterial pneumonia. They provided novel data and demonstrated the activity of DARS2 is controlled by a bacterial-induced ubiquitin E3 ligase subunit, FBXO24. In line with these findings, Fbxo24 knockout mice have elevated DARS2 levels with a robust increase in lung inflammation. In addition, BC-1293 as a FBXO24 inhibitory compound exhibited pro-inflammatory effects both in vitro and in vivo. Although the topic is novel and interesting, several major concerns affect the data interpretation and conclusions of the work.

Major concerns:

Point 1: Fig 1i and 1j, *S. aureus* and *K. pneumoniae* infections were not able to significantly induce FBXO24. The authors stated this may be due to these two strains being gram-positive pathogens. In fact, *K. pneu* is a gram-negative pathogen. The interpretation needs to be carefully revised.

Response 1: We did refer to *K. pneumoniae* as a gram negative in the original manuscript. Regardless, we have included more Ns of *K.pneumoniae* and *S. aureus* and also assessed specificity of responses by evaluating other pathogens. Indeed, unlike *S. aureus*, the bacterial pathogens *K. pneumoniae*, *S. pneumoniae*, and *E. coli* infection all increased FBXO24 protein

expression in BEAS-2B cells or THP-1 cells ((Fig. 1i, Supplementary Fig. 1d,e). These data suggest that FBXO24 levels are variably induced depending on the pathogen or cell type. These results are added to the revised manuscript (Results: page 6, paragraph 1, lines 10-14).

Point 2: Fig 2b, bacterial loads were not significantly reduced in Fbxo24 KO mice. Thus, it is inaccurate to say that “Fbxo24 KO mice showed decreased bacterial loads compared to WT mice” in the manuscript. Also, the sample size mentioned in the figure legend is n= 5-6/group. But only 4 dots were shown in the panel.

Response 2: We have revised this statement to state “a trend toward a decrease” to reflect that these observations approach, but do not achieve statistical significance (Results: page 7, paragraph 1, line 7). Further we have adjusted the figure legend to reflect the accurate range of animals per group.

Point 3: Fig 6g, it is very interesting to see that DARS2 has smaller MW in both sup and exo groups than CL group. It would be better to further discuss the observation.

Response 3: We have added discussion of this observation to the result and discussion sections. In brief, we believe that this smaller fragment may be required for DARS2 secretion or might be a stimulus specific fragment that is secreted in response to infection. Interestingly, cytoplasmic tryptophan-tRNA synthetase (WRS) is secreted as both a full length and “mini” peptide with the former being released in response to infection. Our observed slower and faster migrating bands from immunoblots from supernatant proteins from cells may represent similar stimulus-dependent secretion of cleaved DARS2 fragments that merits further investigation. These comments are added to the revised manuscript (Results: page 13, paragraph 2, lines 13-15) and (Discussion: page 20, paragraph 2, lines 9-13).

Point 4: There are only two biological replicates shown in Fig 1j. Thus, it is not acceptable to perform statistical analysis or make a conclusion.

Response 4: We have repeated additional Ns of this experiment. We have found these results to be consistent with our previous observation the *K. pneumoniae* is induced by FBXO24 infection. Additional Ns have been included in densitometry analysis in Fig. 1j. Multiple experiments and quantification are shown on revised Fig. 1 for *P. aeruginosa* and *K. pneumoniae* by densitometry.

Point 5: Fig 7m-7p, is quite confusing about the procedures for these in vivo experiments. It appears that mice were only treated with DMSO control or BC-1293. But, in the introduction, it was stated that “FBXO24 inhibitor compound, that when administered in experimental pneumonia, displays immunostimulatory activities”. Unfortunately, the authors did not provide any details in the Methods section.

Response 5: We recognize these details require clarification. In these studies, the only treatment was with DMSO as a control or BC-1293 at two doses (2 or 5 µg). We added a section in Methods stating: *In vivo administration of BC-1293*- BC-1293 was prepared in PBS with 5% DMSO with a final concentration of 20 ng/µL and 50 ng/µL. C57BL6 mice were assigned to three experimental groups: DMSO, 2µg or 5µg BC-1293 (*n*=5/group) and received 100µL of respective stocks i.t. Mice were euthanized 16h post-treatment and BAL was collected for differential analysis of immune cells and multiplex ELISA for cytokine concentration (Methods: page 33, paragraph 2, lines 6-11).

Point 6: One big concern comes from the specificity of BC-1293. Since the Fbxo24 KO mice are available, it would be really helpful to test the effect of BC-1293 in Fbxo24 KO mice to validate its target specificity.

Response 6: This is a good idea and opportunity to study drug target validation *in vivo*. First with regard to target specificity, it should be noted that in the prior submission of this manuscript we provided target validation by a protein binding drug titration study (**Fig. 7e**). Given the very low productivity of our breeding pairs of *Fbxo24* knockout mice we were unable to conduct the specific experiment suggested by the reviewer. However, to further validate FBXO24 targeting of BC-1293 to induce cytokine secretion, we utilized precision cut lung slice (PCLS) cultures from *Fbxo24* WT or *Fbxo24* heterozygous (*Fbxo24*^{+/-}) mice. *Fbxo24*-WT PCLS co-treated with Pam3CSK4 (5µg/mL) with increasing concentrations of BC-1293 (0, 1, 10 and 20µM) displayed a significant stimulatory effect of BC-1293 on IL-6 secretion 24h post treatment (new **Fig.7q**). This effect was not observed in PCLS derived from *Fbxo24*^{+/-} mice (new **Fig.7r**) demonstrating that BC-1293 immunostimulatory activity is FBXO24 dependent. These results are added to the revised manuscript (Results: page 17, paragraph 1, lines 14-19 and page 18, paragraph 1, lines 1-2).

Third, we recently we co-treated our sgCon (WT) and *FBXO24* BEAS2B knockout cells (sgO2 (KO) with Pam3CSK4 (200ng/mL) and increasing concentrations of BC-1293 or DMSO control (new **Fig. 7g,h**). Indeed, BC-1293 triggered an increase in cellular DARS2 content after Pam3CSK4 stimulation in the sgCon, but not the *FBXO24* sgO24 cell line. These data indicate under certain circumstances, such as co-stimulation with Pam3CSK4, BC-1293 increases DAR2 abundance in a FBXO24 dependent manner. These results are added to the revised manuscript (Results: page 17, paragraph 1, lines 1-4).

Finally in a related matter regarding drug-target validation *in vitro*, we have had substantial challenges over the last several months regarding drug-target binding affinity assessment. BC-1293 binding affinity requires highly purified and properly folded *Fbxo24* that is

needed for isothermal calorimetry or surface plasmon resonance studies. In our experience purifying E3 ligase components takes time, but also once purified we have found that the purified F-box proteins are in fact often misfolded, wasting time and precious resources. Nevertheless, we embarked on the binding affinity studies the last several months with the first step to purify the E3 ligase subunit. Unfortunately, we encountered several challenges with Fbxo24 purification without success due to issues of adequate purity and yield after cloning of the Fbxo24 ORF and expression in mammalian (Hekexp293) cells. We also considered purchasing recombinant Fbxo24, however, commercial costs are ~20,000 pounds (17K US dollars) for recombinant Fbxo24. Our last experience in purchasing another F-box protein was unfortunate as the purified protein was misfolded and not suitable for binding studies. We would like to pursue Fbxo24-BC-1293 binding using isothermal calorimetry (ITC)/surface-plasmon resonance work in a separate manuscript.

Minor concerns:

Point 7: Fig 2l, it would be better to show Total cells, PMN, or Mono in separate panels to improve the readability.

Response 7: As requested, we adjusted the format of the figure to include three individual graphs (new **Figs 2i-k**).

Point 8: What is HEK293293T cell line? Is it 293 or 293T cells? Please indicate the sources of cell lines used in this study for scientific rigors and reproducibility.

Response 8: This was a typographical error on our part, and it should be HEK293T.

Point 9: The authors used siRNA to knock down DARS2 in Fig. 6d-f. It is inaccurate to use “depletion” when describing the result.

Response 9: We have changed the language to knockdown to reflect that it is a transient effect

using siRNA.

Point 10: More details need to be included in Materials and Methods. For instance, the protocol used for MV, EXO isolation.

Response 10: The isolation protocol and other information have been added to the Materials and Methods section (Results: page 27, paragraph 2, lines 13-20 and page 28, lines 1-3).

Reviewer #2 (Remarks to the Author):

In this manuscript, the authors discovered that the E3 ligase component, FBXO24, is an immunoregulatory protein via the cellular disposal of DARS2, a previously unrecognized secreted protein in human plasma with cytoprotective and immune properties. Additionally, the authors identified an FBXO24 inhibitor compound, that when administered in experimental pneumonia, displays immunostimulatory activities. This is an interesting manuscript, but I still have the following questions.

Point 1: In Figure 1f and 1g, the data source is not clearly indicated, making it unclear whether the results in these two figures are derived from A549 cells or BEAS-2B cells. Similar issues have been found in other parts of the manuscript, which requires a careful re-examination of the original draft, and the experimental methods in the manuscript need to be described in more detail.

Response 1: The figure legends have been corrected to clarify this. **Fig.1f** is a quantification of the representative blot in **Fig.1fe** with additional Ns of BEAS2Bs infected with PA103. Additional information has also been added to several sections of the method section and figure legends have been edited for greater clarity.

Point 2: In the description of Figure 1d-h, "...induction by PA103 infection in A549 and BEAS-

2B cells after 24 h culture. " Curiously, can viable A549 and BEAS-2B cells be harvested after 24 h of PA103 treatment? In addition, BEAS-2B cells were infected with KP and SA for only 6 h, why not the same time as the previous PA103 infection?

Response 2: This was an error on our part and the manuscript **Fig. 1** legend has been corrected appropriately. All three of these experiments were conducted within 6h because longer time points of infection do cause high levels of cell toxicity.

Point 3: In experiments to assess the pattern of polyubiquitylation chain formation, especially for the detection of K48 polyubiquitylation, did the cells use MG-132 treatment? Moreover, as a control experiment, FBXO24 can be added to the experiment in Figure 4i to determine which polyubiquitylation linkage of DARS2 is increased by FBXO24.

Response 3: To optimize our signals on immunoblots, our Co-IPs involved pretreatment of the cells with MG132 [100µM] and Bafilomycin A1 [200nM] to allow accumulation of proteins targeted for degradation prior to cell lysis. This has been added to the revised text (Methods: page 32, paragraph 2, lines 10-12). As requested, we conducted the suggested experiment where ectopically expressed *FBXO24* appears to modulate K33 polyubiquitylation which has been linked in limited studies to lysosomal degradation. We have added these results to **new Supplemental Fig. 4b**. Interestingly, *FBXO24* mediated K33 polyubiquitylation of DARS2 is consistent with our observation that Bafilomycin A1 leads to DARS2 accumulation (**Fig. 4c**).

Point 4: Figure 4 indicates that FBXO24 targets DARS2 for ubiquitylation and degradation, but there seems to be no direct evidence from the experimental results to suggest that DARS2 is degraded through the ubiquitination pathway.

Response 4: We have conducted additional studies using MLN-7424 in BEAS-2B cells and observed that DARS2 levels are induced maximally by using this pan-ubiquitin inhibitor

suggesting degradation through the ubiquitination pathway (**new Fig. 4d**, $n=3$). The results are added to the revised text (Results: page 10, paragraph 2, lines 16-18).

Point 5: The authors mention in Figure 1 " Human transplant rejected lung tissue testing positive in bacterial cultures had a significant increase in FBXO24 mRNA and protein levels compared to uninfected tissue...", and in Figure 6 " The critically-ill infected patients displayed significant increases in DARS2 levels compared to controls ...". "However, the author's other experimental data show that FBXO24 leads to the degradation of DARS2. Are there plausible explanations for these seemingly contradictory results?"

Response 5: This is an astute observation by the reviewer. First it is important to indicate that that these are observations from two distinct and very different data sets. Directly comparing the postmortem samples from 10 deceased individuals is a quite different sample from a prospective longitudinal study of 50 patients in the ICU setting. The differences in sampling could very well be responsible for these seemingly contradictory observations. Nevertheless, while FBXO24 was observed to be induced in transplant-rejected lung samples positive for bacterial infection, we also observed increased plasma levels of DARS2 in infected patients. These results suggest that reduced DARS2 levels after FBXO24-mediated degradation in the lung might trigger a compensatory response in circulation to enhance bacterial clearance. Further studies examining the kinetics of FBXO24 expression and DARS2 abundance in experimental pneumonia and in humans will elucidate the biological association for these proteins in host immunity. This discussion is added to the revised text (Discussion: page 22, paragraph 1, lines 2-9).

Point 6: In Figure 7e, BC-1293 at or above 250 nmol reduced DARS2:FBXO24 binding. Considering the degradation effect of FBXO24 on DARS2, observe whether the content of DARS2 will increase after BC-1293 reduces DARS2:FBXO24 binding. Thus, experiments can

be supplemented: changes in DARS2 content in the presence or absence of BC-1293 in both infected and uninfected conditions.

Response 6: We conducted several follow up experiments to address this question. We first attempted to demonstrate BC-1293 driven accumulation of DARS2 in BEAS2B cells infected with or without PA103. However, these experiments yielded inconsistent data. Next, given our experiments demonstrating that BC-1293 increased DARS2 levels and cytokine secretion in the presence of a potent TLR2/TLR1 ligand, Pam3CSK4, we conducted follow up experiments using this nonbacterial stimulus in combination with BC-1293. We co-treated our sgCon (WT) and *FBXO24* BEAS2B knockout cells (sgO2 (KO) with Pam3CSK4 (200ng/mL) and increasing concentrations of BC-1293 or DMSO control (**new Fig. 7g,h**). Indeed, BC-1293 triggered an increase in cellular DARS2 content after Pam3CSK4 stimulation in the sgCon, but not the *FBXO24* sgO24 cell line. These data indicate under certain circumstances, such as co-stimulation with Pam3CSK4, BC-1293 increases DAR2 abundance in a FBXO24 dependent manner. This discussion is added to the revised text (Results: page 17, paragraph 1, lines 2-4 and Discussion: page 23, paragraph 1, lines 6-8).

Reviewer #3 (Remarks to the Author):

In this study, Johnson et al. revealed that the E3 ligase component, FBXO24, is an immunoregulatory protein via the cellular disposal of DARS2, a previously unrecognized secreted protein in human plasma with cytoprotective and immune properties. The DARS2-FBXO24 interaction mechanism may provide opportunities to modulate host immune responses through small molecule compounds. Overall, this work is a meaningful study with potential for publication. However, there are some key issues that need to be resolved before publication:

(1) Many conclusions in the article lack convincing evidence. The following are just some examples:

Point 1: Figure 1 includes data from only two species, which may not be sufficient to conclusively support the following assertion: Increased FBXO24 protein levels are restricted to gram-negative pathogens. Additional data are required.

Response 1: As requested by the reviewer we examined effects of several other bacterial pathogens using differing cell lines. Interestingly, unlike *S. aureus*, the bacterial pathogens *K. pneumoniae*, *S. pneumoniae*, and *E. coli* infection all increased FBXO24 protein expression in BEAS-2B cells or THP-1 cells (**new Fig. 1i, Supplementary Fig. 1d,e**). These data suggest that FBXO24 levels are variably induced depending on the pathogen or cell type. These results are added to the revised manuscript (Results: page 6, paragraph 1, lines 10-14, Methods: page 30, paragraph 1, lines 1-10).

Point 2: In terms of data significance in Figure 2, Figure 2B lacks significance, thus hampering the derivation of relevant conclusions. Additionally, there are only two values in the KO group in Figure 2E, 2F, and 2G, precluding meaningful significance calculations. The significance markers on heat maps are confusing, such as Figure 2J, some indicators that appear to be significantly different are not actually marked with significance.

Response 2: With regard to bacterial loads in Fig. 2b, we have revised this statement to state “a trend toward a decrease” to reflect that these observations approach, but do not achieve statistical significance (Results: page 7, paragraph 1, line 7). With respect to the lung function studies, we performed additional testing of more mice. *Fbxo24* WT and *Fbxo24* KO mice infected i.n. with PA103 (1×10^5 CFU) displayed no significant difference in lung mechanics (**new Fig. 2l-n**). Here we used a lower amount of bacterial infection in mice because of technical challenges using the

Flexivent system using viable mice to measure lung function. Nevertheless, under these conditions the data suggest that Fbxo24 depletion triggers increased lung innate immune activity that is not associated with significant impairment of lung mechanics. The results are added to the revised text (Results: page 8, paragraph 1, lines 2-10). Last, regarding the cytokine panels, our statistical analysis showing significant differences in the cytokines is indicated with an asterisk; a if not marked as significant it was not found to be significant by statistical analysis.

Point 3: In Figure 6P, the concentration values of various inflammatory factors are too small. The difference between the two groups may be caused by experimental errors. It is recommended to perform additional experimental tests for confirmation.

Response 3: This is a valid concern, as such we have removed the mentioned cytokine data and its discussion from the manuscript. This does not negate BC-1293's immunostimulatory potential in this mouse model as the compound triggers increases in infiltrating immune cells in the BAL and peritoneal fluid.

Point 4: It is appreciated that the authors identified a tool compound, FBXO24 inhibitor BC-1293, through virtual screening. The authors confirmed the efficacy of BC-1293 through various cellular experiments and in vivo activity experiments in mice. However, how did the authors confirm that BC-1293 exerts in vitro and in vivo activity by targeting FBXO24? How is the dosage concentration of BC-1293 determined? Are there any data that can provide information on the binding affinity between BC-1293 and FBXO24?

Response 4: We appreciate the reviewers shared enthusiasm for our novel tool compound and understand that robust target validation is an essential step in compound development. These are all important issues in drug-target validation that over the past year we seriously considered and thus attempted to conduct additional studies in response to the reviewer. First, in the prior

submission of this manuscript we provided cellular target validation by a protein binding drug titration study (**Fig. 7e**). Second, to further validate FBXO24 targeting of BC-1293 to induce cytokine secretion, we utilized precision cut lung slice (PCLS) cultures from *Fbxo24* WT or *Fbxo24* heterozygous (*Fbxo24^{+/-}*) mice (our ability to successfully breed enough *Fbxo24* KO mice [*Fbxo24^{-/-}*] was significantly limited). *Fbxo24*-WT PCLS co-treated with Pam3CSK4 (5µg/mL) with increasing concentrations of BC-1293 (0, 1, 10 and 20µM) displayed a significant stimulatory effect of BC-1293 on IL-6 secretion 24h post treatment (**new Fig.7q**). This effect was not observed in PCLS derived from *Fbxo24^{+/-}* mice (**new Fig.7r**) demonstrating that BC-1293 immunostimulatory activity is FBXO24 dependent. These results are added to the revised manuscript (Results: page 17, paragraph 1, lines 14-19 and page 18, paragraph 1, lines 1-2). Third, we co-treated our sgCon (WT) and *FBXO24* BEAS2B knockout cells (sgO2 (KO) with Pam3CSK4 (200ng/mL) and increasing concentrations of BC-1293 or DMSO control (**new Fig. 7g,h**). Indeed, BC-1293 triggered an increase in cellular DARS2 content after Pam3CSK4 stimulation in the sgCon, but not the *FBXO24* sgO24 cell line. These data indicate under certain circumstances, such as co-stimulation with Pam3CSK4, BC-1293 increases DAR2 abundance in a FBXO24 dependent manner. These results are included in the revised manuscript (Results: page 17, paragraph 1, lines 1-4).

With regard to additional drug-target validation *in vitro*, we have had substantial challenges over the last several months regarding drug-target binding affinity assessment. BC-1293 binding affinity requires highly purified and properly folded *Fbxo24* that is needed for isothermal calorimetry or surface plasmon resonance studies. In our experience purifying E3 ligase components takes time, but also once purified we have found that the purified F-box proteins are in fact often misfolded, wasting time and precious resources. Nevertheless, in

response to the reviewer, we embarked on binding affinity studies the last several months with the first step to purify the E3 ligase subunit. Unfortunately, we encountered several challenges with Fbxo24 purification without success due to issues of adequate purity and yield after cloning of the Fbxo24 ORF and expression in mammalian (HekXP293) cells. We also considered purchasing recombinant Fbxo24, however, commercial costs are ~20,000 pounds (17K US dollars) for recombinant Fbxo24. Our last experience in purchasing another F-box protein was unfortunate as the purified protein was misfolded and not suitable for binding studies. We would like to pursue Fbxo24-BC-1293 binding using isothermal calorimetry (ITC)/surface-plasmon resonance work in a separate manuscript. Finally, the dosage concentration of BC-1293 was determined by *in vitro* and animal testing in pilot experiments to detect an effect of the compound on efficacy.

Point 5: The concentration units in the article should be standardized, for example, ug/uM should be corrected to $\mu\text{g}/\mu\text{M}$.

Response 5: This has been corrected throughout the revised text.

REVIEWERS' COMMENTS

Reviewer #1 (Remarks to the Author):

All my concerns have been addressed satisfactorily by adding additional panels and information. The quality of the work has been significantly improved and no further weakness is noted.

Reviewer #2 (Remarks to the Author):

I focused only on my review and the points that I have raised. All these points have been properly addressed and the revised manuscript can be accepted in my opinion for publication.

Reviewer #3 (Remarks to the Author):

The authors have made commendable efforts in revising the manuscript to address previously raised issues, thus improving the quality of the manuscript. However, there is no significant difference in the data in Figure 2b, and drawing conclusions based on non-statistically significant data may weaken the convincingness of the conclusion. Additionally, the differences in the results presented in the new Figure 2l-n compared with the previous version demonstrate the importance of sufficient sample size and data significance to ensure the accuracy and credibility of the conclusions. Overall, while the manuscript has shown improvement, these concerns highlight the need to carefully review all conclusions affected by sample size and statistical significance.